# Short-term variation of pH in seawaters around coastal areas of Japan: Characteristics and forcings

Tsuneo Ono[1], Daisuke Muraoka[2], Masahiro Hayashi[3], Makiko Yorifuji[3*], Akihiro Dazai[4], Shigeyuki Omoto[5], Takehiro Tanaka[6], Tomohiro Okamura[7], Goh Onitsuka[7], Kenji Sudo[7], Masahiko Fujii[8#], Ryuji Hamanoue[8], and Masahide Wakita[9]

[1]Fisheries Resources Institute, Japan Fisheries Research and Education Agency, Yokohama, 236-8648, Japan
[2]Fisheries Technology Institute, Japan Fisheries Research and Education Agency, Miyako, 027-0097, Japan
[3]Marine Ecology Research Institute, Kashiwazaki, 945-0017, Japan
[4]Center for Sustainable Society, Minami-Sanriku, 986-0775, Japan
[5]Eight-Japan Engineering Consultants Inc., Okayama, 700-8617, Japan
[6]Satoumi Research Institute, Okayama, 704-8194, Japan
[7]Fisheries Technology Institute, Japan Fisheries Research and Education Agency, Hatsukaichi, 739-0452, Japan
[8]Faculty of Environmental Earth Science, Hokkaido University, Sapporo, 060-0810, Japan
[9]Mutsu Institute for Oceanography, Japan Agency for Marine-Earth Science and Technology, Mutsu, 035-0022, Japan
*now at National Institute of Advanced Industrial Science and Technology, Tsukuba, 305-8567, Japan
#now at Otsuchi Coastal Research Center, Atmosphere and Ocean Research Institute, the University of Tokyo, Otsuchi, 028-1102, Japan

*Correspondence to*: Tsuneo Ono (tono@fra.affrc.go.jp)

**Abstract.** The pH of coastal seawater varies based on several local forcings, such as water circulation, terrestrial inputs, and biological processes, and these forcings are changing along with global climate change. Understanding the mechanism of pH variation in each coastal area is thus important for a realistic future projection that considers changes in these forcings. From 2020 to 2021, we performed parallel year-round observations of pH and related ocean parameters at five stations around the Japanese coast (Miyako Bay, Shizugawa Bay, Kashiwazaki Coast, Hinase Archipelago, and Ohno Strait) to understand the characteristics of short-term pH variations and their forcings. Annual variability (~1 standard deviation) of pH and aragonite saturation state ($\Omega_{ar}$) were 0.05–0.09 and 0.25–0.29, respectively, for three areas with low anthropogenic pressures (Miyako Bay, Kashiwazaki Coast, and Shizugawa Bay), while it increased to 0.16–0.21 and 0.52–0.58, respectively, in two areas with medium anthropogenic pressures (Hinase Archipelago and Ohno Strait in Seto Inland Sea). Statistical assessment of temporal variability at various timescales revealed that most of the annual variabilities in both pH and $\Omega_{ar}$ were derived by short-term variation at a timescale of < 10 days, rather than seasonal-scale variation. Our analyses further illustrated that most of the short-term pH variation was caused by biological processes, while both thermodynamic and biological processes equally contributed to the temporal variation in $\Omega_{ar}$. The observed results showed that short-term acidification with $\Omega_{ar}$ < 1.5 occurred occasionally in Miyako and Shizugawa Bays, while it occurred frequently in the Hinase Archipelago and Ohno Strait. Most of such short-term acidified events were related to short-term low-salinity events. Our analyses showed that the amplitude of short-term pH

variation was linearly correlated with that of short-term salinity variation, and its regression coefficient at the time of high freshwater input was positively correlated with the nutrient concentration of the main river that flows into the coastal area.

## 1 Introduction

The ocean is witnessing a reduction in its pH due to anthropogenic $CO_2$ sequestration both in open (e.g., Bates et al., 2014; Iida et al., 2021; Jiang et al., 2019; Lauvset et al., 2015; Takahashi et al., 2014) and coastal oceans (e.g., Carstensen and Duarte, 2019; Duarte et al., 2013; Hauri et al., 2013; Ishizu et al., 2019; Ishida et al., 2021; Yao et al., 2022). In the coastal ocean, pH shows short time variation caused by several processes such as water mass changes (e.g., Johnson et al., 2013; Ko et al., 2016; Wakita et al., 2021), coastal upwelling (e.g., Barton et al., 2012; Booth et al., 2012; Feely et al., 2008, 2016; Vargas et al., 2015), rivers inputs (e.g., Cai et al., 2017; Fujii et al., 2021; Gomez et al., 2021; Salisbury and Jönsson, 2018; Salisbury et al., 2008), terrestrial nutrients' inputs (e.g., Cai et al., 2011; Guo et al., 2021; Kessouri et al., 2021; Provoost et al., 2010; Sunda and Cai, 2012; Wallace et al., 2014), and various coastal biological processes (e.g., Delille et al., 2009; Mongin et al., 2016; Lowe et al., 2019; Ricart et al., 2021; Yamamoto-Kawai et al., 2021). The amplitude of short-term pH variation often exceeds that of the decadal-scale long-term pH trend, and hence, blurs the signal of anthropogenic $CO_2$-induced acidification of coastal seawater (e.g., Borges and Gypens, 2010; Duarte et al., 2013, Johnson et al., 2013; Provoost et al., 2010; Salisbury and Jönsson, 2018). Short-term pH variations in coastal waters are important for local ecosystems as they are mostly caused by natural forcings that have been acting before the industrial period, and hence, the local ecosystem is expected to adapt to such short-term pH variations as long as they are natural in terms of timing and amplitude. For example, *Ostrea lurida*, a native oyster in Netarts Bay, Oregon, USA, has adjusted its spawning season before and after the summer upwelling season so that its larvae can avoid low-pH waters (Waldbusser et al., 2014). Several anthropogenic perturbations, such as changes in land use, sewerage treatment, vanishment of seagrass bed, and modification of coastal topography, can change these forcings, shifting the timing and/or amplitude of natural short-term pH variation (e.g., Papalexiou and Montanari, 2019; Hoshiba et al., 2021). Understanding the present situation as well as the mechanism of short-term pH variation in coastal waters is thus critical for evaluating the risk of acidification in coastal areas.

Japan consists of 14,125 islands distributed in a wide latitudinal range from 20 °N to 45 °N in the western North Pacific, containing diverse coastal environments from coral reefs to seasonal floating sea ice. Japan is a highly developed country, and a significant portion of its coastal area has experienced various types of anthropogenic perturbations. The country's Ministry of the Environment (MOE) conducts regular pH monitoring at over 2,000 coastal stations around Japan from the early 1980s until the present, and the obtained data showed significant variability in the multi-decadal pH trend from $-0.012 \text{ y}^{-1}$ to $+0.009 \text{ y}^{-1}$ throughout the stations (Ishizu et al., 2019). The observed range of pH trend within the Japanese coast is equivalent to 85% of that observed in 83 coastal systems in the world (Carstensen and Duarte, 2019), and this result suggests that the Japanese coastal area can be considered as a "sample shelf" for coastal acidification studies. The MOE monitors pH

monthly and seasonally, and several of pH stations of MOE, especially those in northern areas, lack wintertime observations (Ishizu et al., 2019). We thus need additional pH observations with a higher time resolution to understand the characteristics of short-term pH variation at a timescale of < 30 days around the Japanese coast to assess its variability and mechanisms. A number of scientists have already started such pH monitoring in coastal stations in Japan (e.g., Christian and Ono, 2019; Fujii et al., 2021, 2023; Ishida et al., 2021; Wakita et al., 2021). However, most of these observations were started recently and the

summarisation of observed results among these stations is yet to be made. In this study, we carried out the first synthesis effort of such high-resolution pH monitoring stations along coastal areas in Japan that are operated by different founders/programs, by summarising continuous monitoring data of pH observed during 2020–2021 at five stations around the Japanese coast. Here, we describe and discuss the amplitude of pH variation in each timescale, similarity, and dissimilarity in the characteristics of variation, and their forcings.

## 2 Observations and settings of study areas

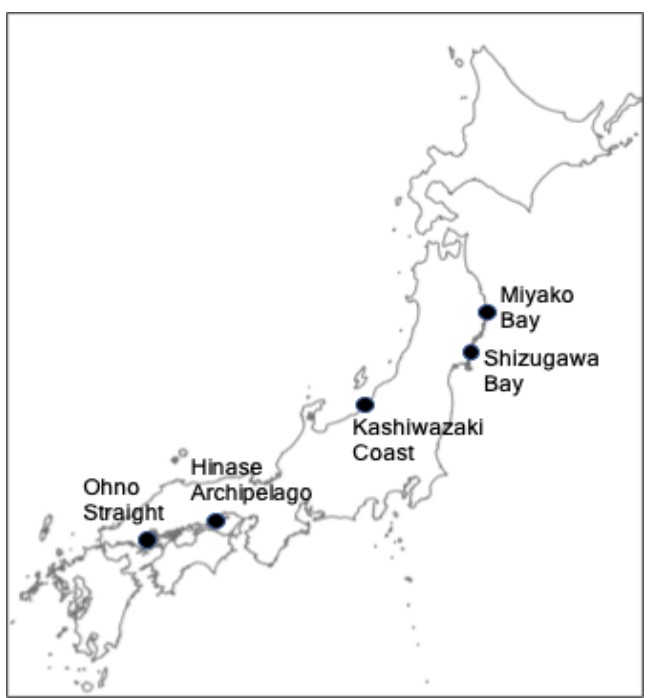

**Figure 1: Map of the five study stations along the coast of Japan.**

Hydrographic monitoring, including pH monitoring, was performed at five stations around the coast of Japan in the following stations, Miyako Bay, Shizugawa Bay, Kashiwazaki Coast, Hinase Archipelago, and Ohno Strait (Fig. 1), from 2020 to 2021. Miyako Bay and Kashiwazaki Coast were selected to represent coastal environment with relatively low anthropogenic nutrient loadings, while the other three stations were selected from major farming areas of pacific oyster. The detailed settings of the areas and observation procedures are described in the following sections.

### 2.1 Miyako Bay

Miyako Bay is located in the northern part of Honshu Island facing the western North Pacific, with a bay area of 24 km$^2$ and 4.8-km wide bay mouth (Fig. 2(a)). The outer bay area is usually occupied by temperate western North Pacific water, which brings enough nutrients in winter to support seaweed farms in the coastal area (Kakehi et al., 2018). Additional nutrients to the bay are provided by three rivers: Hei, Tsugaruishi,

and Tashiro (25.8, 5.16, and 3.78 m$^3$ s$^{-1}$, respectively, for annual average flow rate; Okada et al., 2014), although the quantity of nutrients input by the rivers are limited to low levels (57, 11, and 8 tN y$^{-1}$ for Hei, Tsugaruishi, and Tashiro Rivers, respectively; Bernardo et al., 2023). With a population of 60,000 residents within the hinterland, Miyako Bay has maintained good water quality with 1–2 mg L$^{-1}$ of chemical oxygen demand (MOE, 2022). While kelp beds are well developed near the shoreline, wakame seaweed (*Undaria pinnatifida*) farmyards are developed in the middle of the bay.

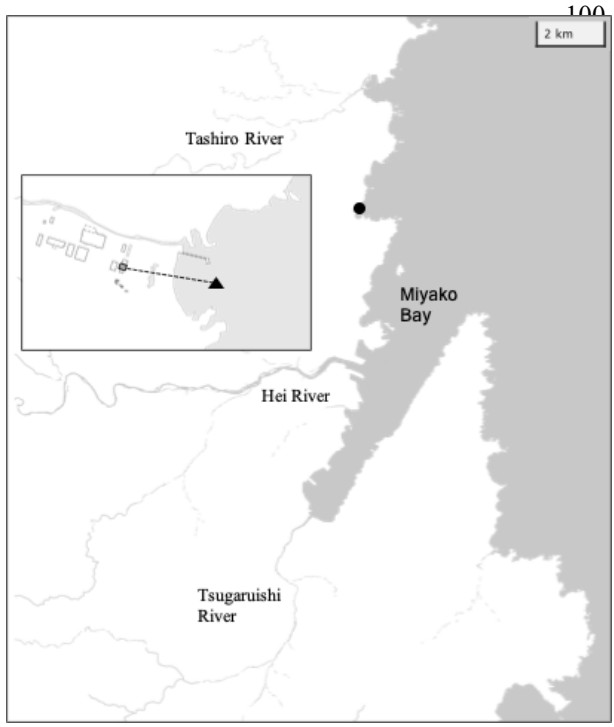

**Figure 2: Map of Miyako Bay with the location of the station (solid circle). Overlayed map shows detailed structure of the station. Grey square and solid triangle represent locations of settling tank and water intake, respectively.**

The monitoring site was located in front of the Miyako Field Station of the Japan Fisheries Research and Education Agency, which is located north of the bay mouth (141° 58' 5" E and 39° 41' 28" N; Fig 2). This area frequently encounters severe winter storms, so we set the monitoring station in the settling tank (3.6 × 2.8 × 5.4 m) of the field station, in which coastal water is continuously pumped from the water intake located 200 m off the coastline and 1–3 m in depth (Fig. 2). Sensors for pH (Kimoto Electronic, SPS-14), dissolved oxygen (DO) (JFE Advantech, AROW2), and salinity/water temperature (JFE Advantech, ACTW) were then moored at a depth of 1 m in the settling tank. Mechanical precision of each sensor is ±0.003 pH units for pH, ±2% for DO, ±0.008 for salinity and ±0.01 ºC for water temperature, respectively. The difference in pH between water intake and settling tank was measured for two weeks during the monitoring period, and it was confirmed that the difference in pH between the water intake and settling tank was < 0.006 pH units. The sampling frequency of each sensor was set to 1 h. All sensors were replaced every 2 months because of the limitation of batteries, and the DO sensor was calibrated by air-saturated pure water and sodium sulphite

solution, while the pH sensor was calibrated against tris(hydroxymethyl)aminomethane and 2-amino-2-methyl-1-propanol-buffered artificial seawaters provided by FUJIFILM Wako Pure Chemical Corporation (Cat. No. 017-28191 and 010-28181, respectively) at the beginning of each deployment. Along with these measurements, discrete water samples were taken from the tank at a depth of 1 m using a 1.5-L Niskin bottle every week. Subsamples for salinity, nutrients, dissolved inorganic carbon (DIC), and total alkalinity (TA) were then taken

and stored. Salinity and nutrients were measured at the Yokohama Laboratory of the Japan Fisheries Research and Education Agency using a salinometer (Guildline Instruments 8400 B) and continuous flow analyser (Seal Analytical QuAAtro 39), respectively, while DIC and TA were measured at the Mutsu Institute for Oceanography of the Japan Agency for Marine-Earth

Science and Technology (MIO-JAMSTEC) using a carbon coulometer (UIC  CM 5012 with Nippon ANS MODEL 3000A) and an open-cell titrator (Kimoto Electric ATT-15) calibrated against the seawater reference materials provided by KANSO

Corporation (Wakita et al., 2021). Measurement precision was ±1 μmol kg$^{-1}$ for both DIC and TA. The pH of the seawater at the time of each sampling (pH$_{discrete}$) was calculated from DIC and TA using the CO2sys program v2.1 (Pierrot and Wallace, 2006), with the settings of Lueker et al. (2000) for the dissociation constant of carbonate, Dickson (1990) for the dissociation constant of bisulphate, and Lee et al. (2010) for the aqueous boron concentration. The drift of the pH sensor during deployment was corrected based on the difference between pH$_{discrete}$ and the pH measured by the sensor. The relationship between the

measured TA and salinity was evaluated as a linear function, and the time series of TA throughout the sensor deployment was calculated based on the salinity–TA relationship. The saturation states of calcite ($\Omega_{ca}$) and aragonite ($\Omega_{ar}$) were then calculated from water temperature, salinity, nutrient concentrations, DIC, and TA using the CO2sys program. This monitoring program, founded by the Study Biological Effects of Acidification and Hypoxia (BEACH) of the Environment Research and Technology Development Fund of the Environmental Restoration and Conservation Agency, which started on 1 July 2020 and ended on

21 September 2021.

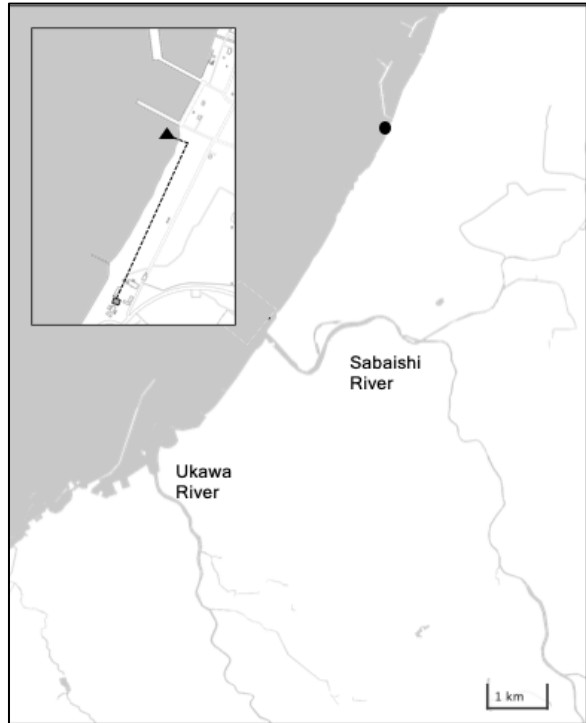

**Figure 3: (a) Detailed map of Kashiwazaki Coast with the location of the station (solid circle). (b) Locations of water intake and settling tank in Kashiwazaki Observatory.**

## 2.2 Kashiwazaki Coast

Kashiwazaki City is located at the centre of Honshu Island facing the Sea of Japan. Its coastline has little flexure, with the periodic occurrence of shallow sandy beaches and small rocky reefs (Fig. 3(a)). While benthic biomass is quite low in sandy beach coastal areas, local *Sargassum* seaweed beds are formed off rocky reefs. The low-nutrient Tsushima Warm Current flows throughout the year off the narrow band of low-salinity coastal water (Niigata Prefectural Institute for Fisheries and Oceanography, 1998). Kashiwazaki City has a population of 79,000 with a cultivated land area of 4,890 ha (mainly rice paddy fields) and ~200 plants of manufacturing industries. Sewage waters from these civil activities are released into coastal areas via two small rivers, Sabaishi and Ukawa Rivers, which bring low salinity and nutrients. The quality of off-Kashiwazaki coastal waters has been well maintained, with a chemical oxygen demand of < 2 mg L$^{-1}$ (MOE, 2022). However, 30 years of monitoring by the Marine Ecology Research Institute shows that

the pH of off-Kashiwazaki coastal water has decreased at a rate of $-0.003$ y$^{-1}$ because of the increase in water temperature and
160 concentration of atmospheric $CO_2$ (Ishida et al., 2021).

Similar to the Miyako site, sensors for pH, DO, and salinity/water temperature were set within the water tank (3,000 L), to which coastal water was continuously pumped from the water intake set in front of the Marine Ecology Research Institute (138 °35' 24″ E and 37 °25' 30″ N, Fig. 3(b)). The settings of each sensor, water sampling, and measurements were the same as those of the Miyako site. This monitoring program, also founded by BEACH, started on 16 July 2020 and ended on 24
August 2021.

## 2.3 Shizugawa Bay

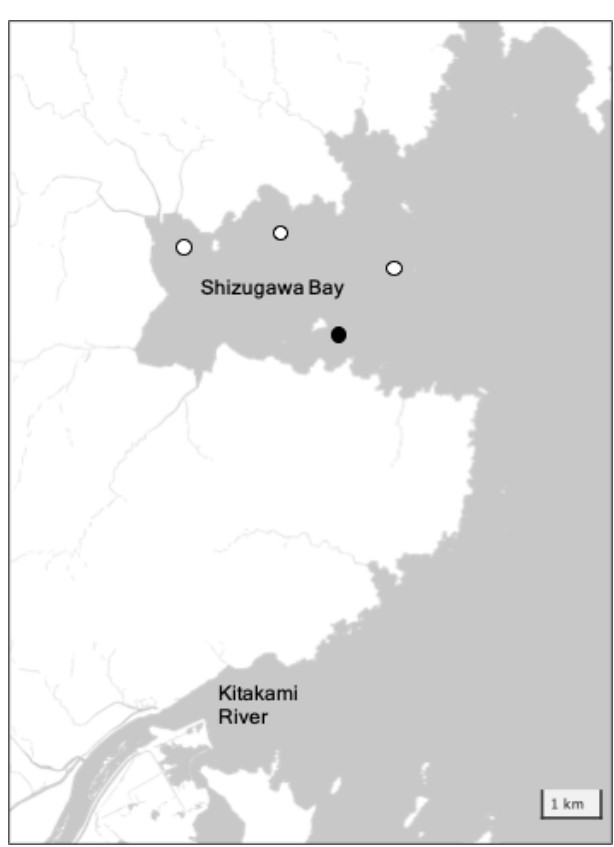

**Figure 4: Detailed map of Shizugawa Bay with the location of the station. Open circles represent stations installed by the Ocean Acidification Adaptation Project. Solid circle represents the location of station S3 used in this study.**

Shizugawa Bay is located ~100 km south of Miyako Bay, with a bay area of 46.8 km$^2$ and a 6.6-km wide bay mouth (Fig. 4). The Oyashio-oriented cold-water and Kuroshio-oriented warm-water masses intermittently occupy the outer bay, and the former water brings high quantities of nutrients to this bay in winter, similar to that in the Miyako Bay. The large Kitakami River (water transport of 390 m$^3$ s$^{-1}$), flows out to the sea south of Shizugawa Bay, bringing nutrients of $6 \times 10^3$ tN y$^{-1}$ to the coastal area (Sugimura et al., 2015). In addition, 10 small rivers flow directly into the bay, and the total nutrient flux provided by these rivers was estimated at 270 tN y$^{-1}$ (Yamamoto et al., 2018). Shizugawa Bay has been widely used for the aquaculture industry, especially for culturing Pacific oysters and silver salmon. Historically, the extent of aquaculture utilisation occasionally becomes too heavy, resulting in the emergence of low-oxygen deep waters in the bay caused by the degradation of organic materials derived from aquaculture. After the Great East Japan Earthquake in 2011, oyster farmers in Shizugawa Bay decided to reduce the number of their oyster rafts to one third of what they had before the earthquake to reduce the environmental impact of aquaculture on the bay ecosystem, which was on the way to recovery from the tsunami disaster. As a result, the DO of the bottom water in the bay showed a remarkable increase to > 1 mg L$^{-1}$ (Yamamoto et al., 2017).

Since 2020, hydrographic conditions, including pH, have been observed at four stations within Shizugawa Bay (Fig. 4) by the Ocean Acidification Adaptation Project (OAAP) founded by the Nippon Foundation (Fujii et al., 2023). Detailed settings of the observations are described in Fujii et al. (2023), and here, we reproduce its brief outline: Sensors for water temperature, salinity, DO, and pH were set at a depth of 1 m in each station to collect data at a temporal resolution of 1 h. Water samples for salinity, nutrients, DIC, and TA were taken from each station monthly, and the sampling interval was enhanced to every 15 days during summer. Salinity, nutrients, DIC, and TA were measured by the same methods as for the Miyako site, although measurements were made at the Kesen-numa Laboratory of Miyagi Prefecture Institute for Fisheries Sciences and MIO-JAMSTEC for nutrients and other properties, respectively. Similar to the Miyako site, pH$_{discrete}$ was calculated from DIC and TA and used for the drift correction of the pH sensors. The time series of TA throughout the sensor deployment was also calculated based on the observed TA–salinity relationship. In this study, we used data from Station S3 (Fig.4) as a representative of Shizugawa Bay, as this station is located in oyster farming areas, the largest aquaculture industry, which is the largest source of organic carbon in this bay. The measurement of water temperature, salinity, and pH was started on 20 August 2020, while that of DO on 27 April 2021. All parameters are continuously measured till date; however, we use data from August 2020 to December 2021 to maintain synchronicity with data of other stations.

## 2.4 Hinase Archipelago

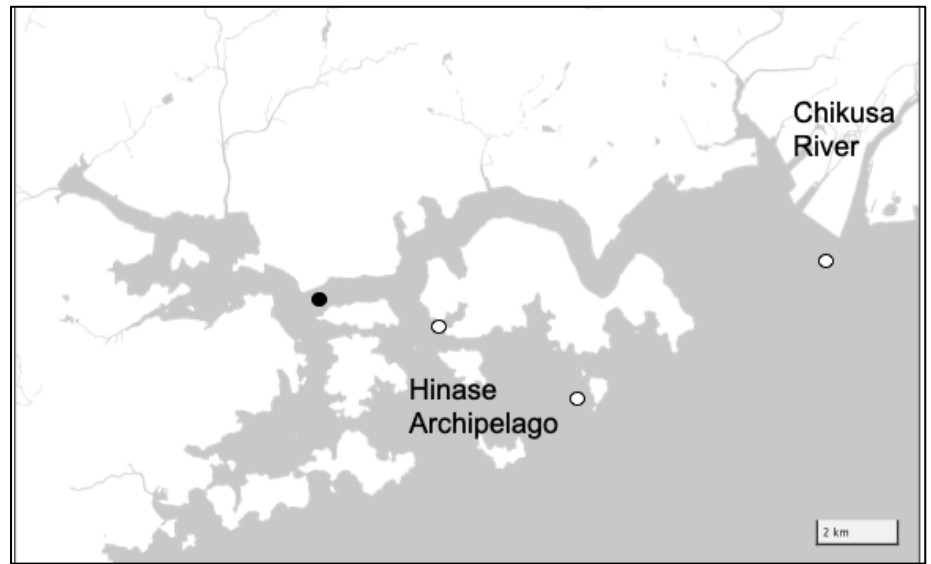

**Figure 5: Detailed map of Hinase Archipelago with the location of the station. Open circles represent stations installed by the Ocean Acidification Adaptation Project. Solid circle represents the location of station H2 used in this study.**

The Seto Inland Sea is the largest inner sea in Japan (~23,200 km$^2$) surrounded by the Honshu, Shikoku, and Kyushu Islands, and the Hinase Archipelago is located in the middle of the Seto Inland Sea. This area consists of four major waterways and many small canals divided by Honshu Island and eight other small islands with a water depth of ~6 m (Fig. 5). The Seto Inland Sea is a moderately eutrophic area, receiving 53.0 km$^3$ y$^{-1}$ of freshwater and 141,000 tN y$^{-1}$ of nitrate from the surrounding lands (Higashi et al., 2018; Nishijima, 2018). The water

exchange rate between the Seto Inland Sea and the outer North Pacific is also high (residence time of ~8 months; Yanagi and Ishii, 2004), and as a result, the nutrient concentration of surface water in the Seto Inland Sea is maintained at a moderate level

(~0.4 μM of DIN near the Hinase Archipelago; Tsukamoto and Yanagi, 1998). Historically, nutrient loading was the highest in the 1980s (~235,000 tN y$^{-1}$; Nishijima, 2018). While the long-standing efforts of local communities succeeded in reducing it to the current level, significant quantities of organic nutrients still remained in the bottom sediment, creating local internal sources of nutrients in the Seto Inland Sea (e.g., Yamamoto et al., 2021). Although there is no large point source of nutrients, such as chemical plants, near the Hinase Archipelago, this area also receives nutrients from the Bizen city, that has a population

of 36,000 residents. The Chikusa River, with an average flow rate of 33.5 m$^3$ s$^{-1}$, flows in the eastern edge of this area, while several other small streams provide additional freshwater. The coastal area of the southern Hinase Archipelago is filled with eelgrass beds, while the northern coastal areas are partially filled with artificial shorelines, such as port facilities and breakwaters. Oyster aquaculture is widely used in open-water areas. The OAAP also launched four stations in this area (Fig. 5). Settings of sensors, water sampling, and its measurements were the same as those of Shizugawa Bay, with the exception

that nutrient measurements were done by the Okayama Prefectural Institute of Fisheries Sciences (see Fujii et al., 2023 for details). In this study, we used data from Station H2 (Fig.5) to represent the Hinase Archipelago, as this station is located in the centre of oyster farming areas. Measurements of water temperature, salinity, and pH were started on 29 August 2020, while measurements of DO was started on 10 June 2021. All parameters are continuously measured till date; however, we use data from August 2020 to December 2021.


## 2.5 Ohno Strait

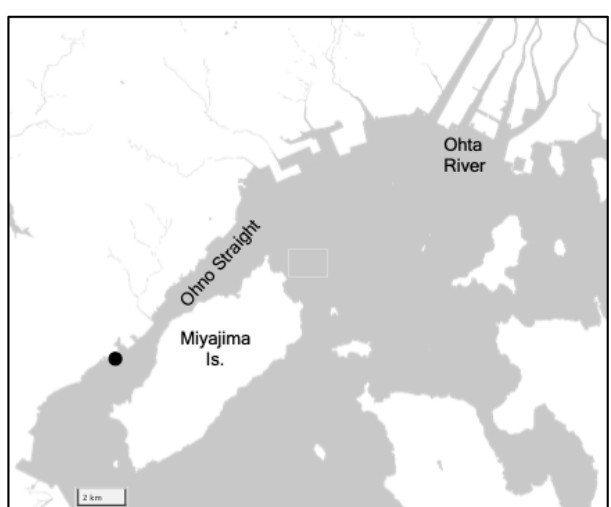

**Figure 6: Detailed map of Ohno Strait with the location of the station (solid circle).**

Ohno Strait is the small strait between Honshu and Miyajima Islands, which are located at the southern boundary of Hiroshima Bay in the western Seto Inland Sea (Fig. 6). As part of the Seto Inland Sea, the surface water surrounding Ohno Strait is moderately eutrophic (~2 μM of DIN; Tsukamoto and Yanagi, 1998), although it is slightly lower than the Hinase Archipelago, reflecting an east–west gradient of surface nutrient concentration. The Ohta River, which flows through Hiroshima city with a population of 1.2 million, adds 1,930 tN y$^{-1}$ of nitrogen into Hiroshima Bay (Yamamoto et al., 2002). Part of the Ohta River water plume flows into the Ohno Strait (Abo and Onitsuka, 2019), providing nutrients to the surface layer of this strait. Industrial areas with a population of 115,000 are developed along the western coast of this strait,

providing another nutrient source to this area. The strait itself is lightly used for oyster farming, and the bottom sediment in the strait is mainly filled by silt and shell fragments.

The monitoring station was launched in the experimental raft off the Hatsukaichi Field Station of the Japan Fisheries Research and Education Agency, with a water depth of 3–6 m depending on the tide. A set of sensors (Kimoto Electronic SPS-14 for pH, JFE Advantech AROW2 for DO, and JFE Advantech ACTW for salinity/water temperature) were fixed from the raft at a
depth of 1 m, and each parameter was measured at an interval of 1 h. The methods and frequencies for sensor calibration and discrete water sampling were the same as those of the Miyako site. This station was launched on 22 June 2021 by the OAAP and is continuing to operate till date; however, we used data from June 2021 to December 2021.

## 3 Results

### 3.1 Temporal variation of water temperature, salinity, and DO


Figure 7 shows the time series of water temperature, salinity, and DO at the five stations. All stations showed similar seasonal variations in water temperature, with the highest and lowest temperatures in July and February, respectively (Fig. 7(a)). The seasonal amplitude was the highest in Hinase and the lowest in Miyako. At all stations, water temperature showed significant day-to-day variation with a timescale of < 10 days, and also diurnal variations. We discuss this further in Section
4.1.

Time series of salinity in Shizugawa, Hinase, and Ohno show similar patterns of seasonal variation, low in summer/autumn and high in winter (Fig. 7(b)), suggesting that the main source of freshwater in these areas are rainfall events (during the rainy season in June/July and typhoon season in August–October). In contrast, in Kashiwazaki, salinity is high in summer and low in winter, suggesting that the main freshwater source in this area is snowfall in winter. The salinity of the
Miyako site shows two low-salinity peaks, one in spring and the other in autumn, suggesting that this area is affected by both snowmelt waters and typhoon events (JFA, 2004). In Miyako, Shizugawa, Hinase, and Ohno, where the freshwater input is significant in summer/autumn, salinity frequently showed short-term drawdown that is synchronised with local rainfall events. The amplitude of sporadic salinity drawdown was extremely high in Hinase and Ohno, where the surface salinity temporally reached to less than 10 (Fig. 7(b)). Amplitude of seasonal annual variation in these two areas reaches over 25, which
corresponds to the largest annual salinity amplitude among the worlds' 83 stations collected in Carstensen and Duarte (2019). We discuss the effect of these short-term low salinity events on pH in Section 4.3.

DO showed similar patterns of seasonal variation: low in summer and high in winter, although the durations were short in Hinase and Ohno (Fig. 7 (c)). This pattern suggests that the seasonal variation of DO is mainly driven by variation of oxygen solubility induced by water temperature rather than biological processes (see Section 4.2 for more details). This
seasonal variation overlaps with day-to-day variation with a duration of ~10 days, and the amplitude of such day-to-day

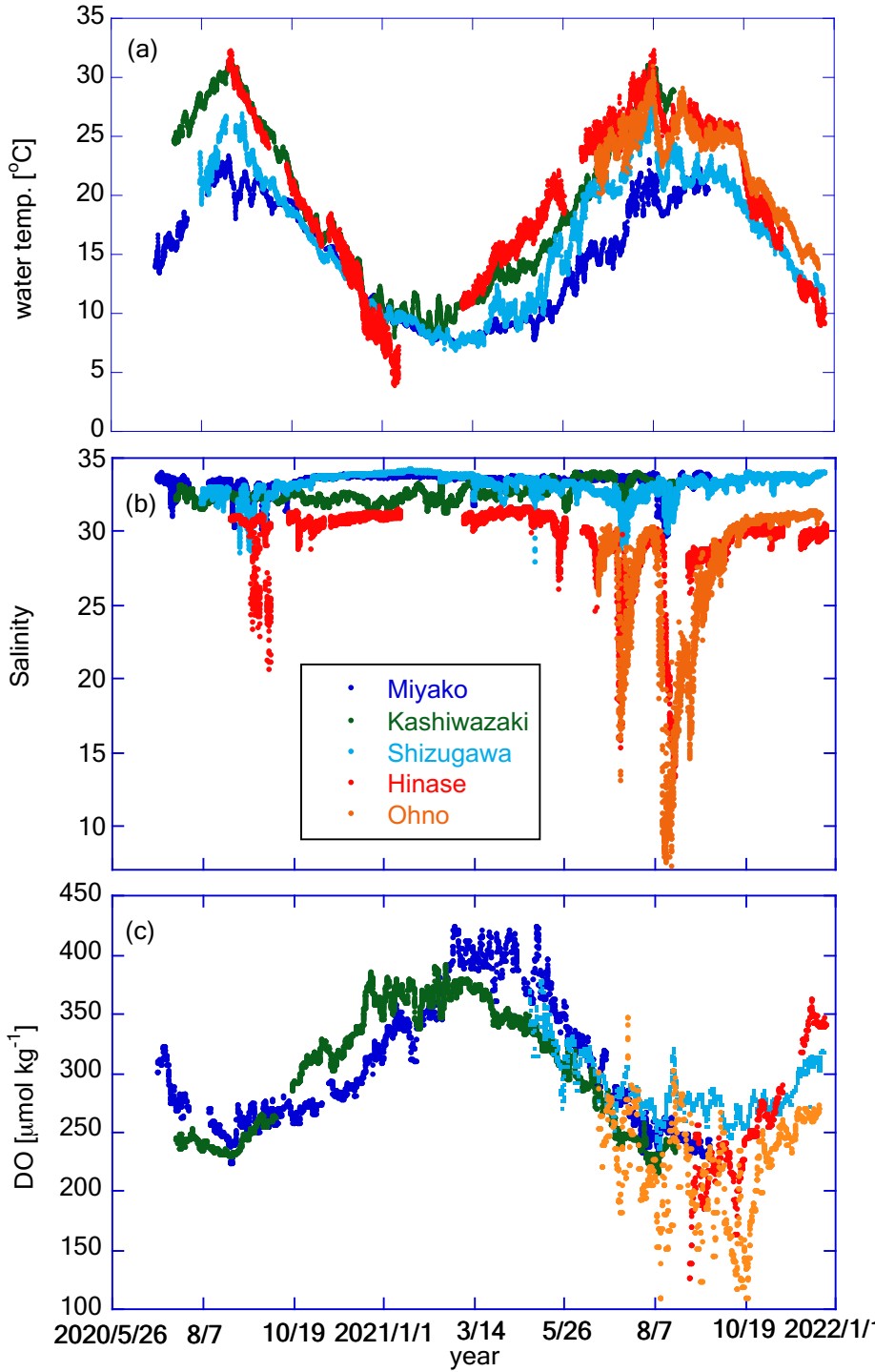

**Figure 7: Time series of (a) water temperature, (b) salinity, and (c) DO in the five stations. Legends of colour plots are the same for all panels as shown in Fig. 7(b).**

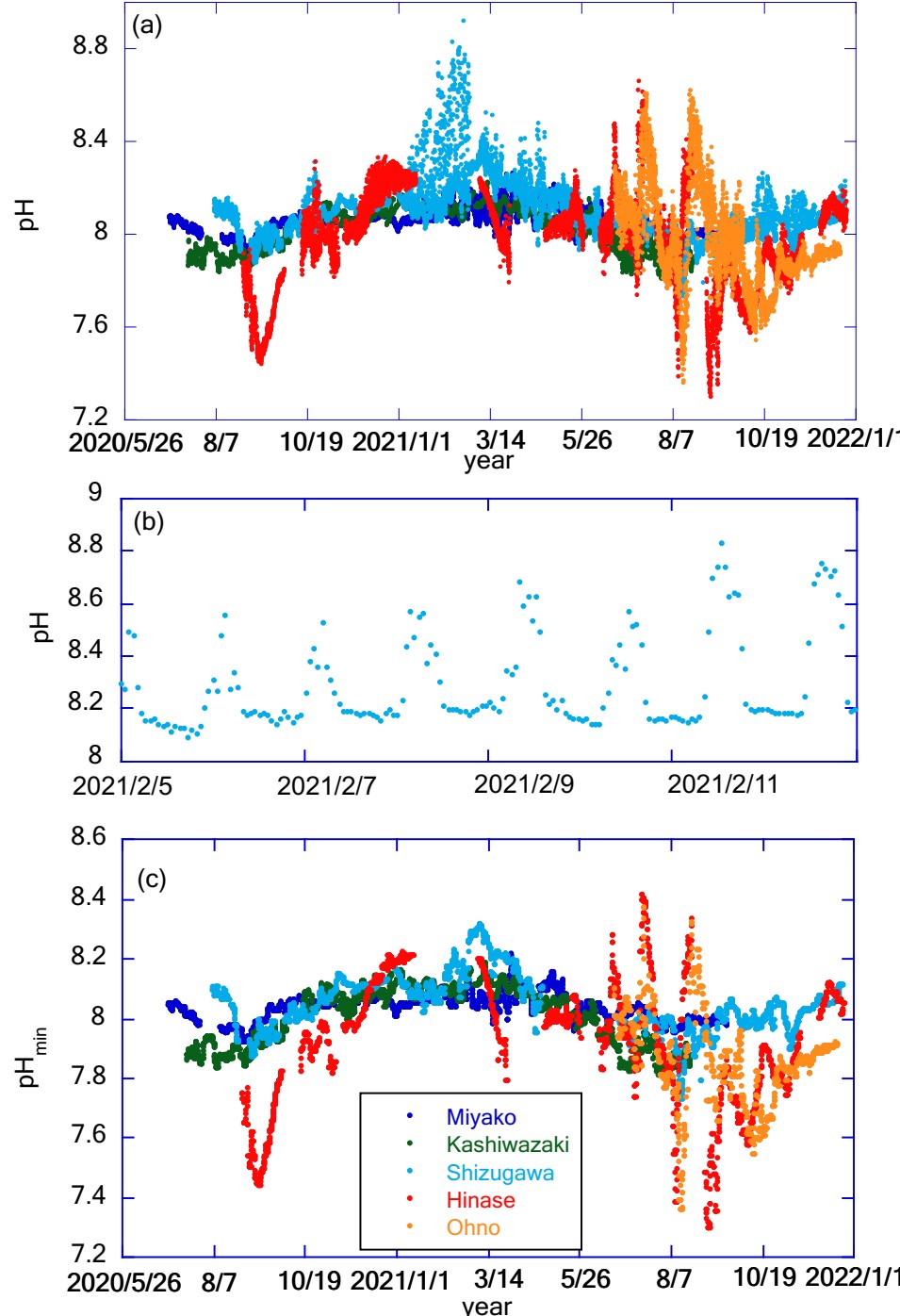

Figure 8: (a) Time series of pH in each station. (b) An example of detailed pH time series. Data in Shizugawa Bay from 2 to 9 February 2021 are presented here. (c) Time series of $pH_{min}$ in each station. Legends of colour plots are the same for all panels as shown in Fig. 8(c).

variation is especially significant in Hinase and Ohno in summer and autumn. Because of this short-term day-to-day variation, DO in Hinase and Ohno were occasionally below 150 μmol kg$^{-1}$ in summer and autumn, indicating that water conditions in these two areas occasionally become significantly undersaturated in DO, even in surface waters.

## 3.2 Temporal variation of pH

Figure 8(a) shows the time series of pH after drift correction at five stations (hereafter, all pH values are reported in total scale in this paper). Although the mechanistic precision of the pH sensor is ±0.003 pH units, the uncertainty of the pH value measured by the glass electrode is mainly controlled by the precision of its drift correction. In the case of Miyako, where drift correction was performed weekly, we set two pH sensors in the same settling tank for two weeks to evaluate the reliability of pH data after the drift correction process. Values of pH obtained from the two sensors matched with that of 2 standard deviations (SD) of ±0.010 pH units, and we used this value as the uncertainty of pH values obtained at Miyako, Kashiwazaki, and Ohno stations. In Shizugawa and Hinase, drift corrections were made at longer intervals, and Fujii et al. (2023) evaluated the uncertainty of pH at these two stations at ±0.015 pH units. Figure 8(a) shows significant daily fluctuations far higher than these pH uncertainties. The diurnal amplitude was the highest in winter in Shizugawa Bay, where the difference in pH between daytime and night-time exceeded 0.8 pH units. High diurnal variations in pH and/or $pCO_2$ are often measured in shallow coastal ecosystems (e.g., Waldbusser et al., 2014; Fujii et al., 2021; Ricart et al., 2021). In most cases, high diurnal pH variation is observed in summer, and the highest diurnal amplitude of pH in winter has rarely been reported. Such patterns raise the possibility of biofouling in pH sensors (e.g., Venkatesan et al., 2017), although visual inspection at the time of sensor exchange did not show significant adherence of biomes during winter. However, the detailed pH variation in each day (Fig. 8(b)) showed that during the night-time it was relatively low and constant, even during significant diurnal variation. This is probably because the organic material produced by the fouling biomes during daytime had settled down from the sensors, and hence, the effect of the decomposition of organic materials during night-time remained low even in the high diurnal variation period. We do not further discuss diurnal variations of pH, as we do not have definite evidence for the effect of biofouling in winter in Shizugawa Bay. We alternatively calculated the daily minimum pH (pH$_{min}$), that is usually observed at night, and analysed the day-to-day variation of pH$_{min}$ at various timescales (Fig. 8(c)). Several rearing experiments have suggested that coastal organisms are affected by pH$_{min}$ rather than the daily average pH (e.g., Onitsuka et al., 2018), therefore, the daily variation of pH$_{min}$ was analysed instead of pH.

pH$_{min}$ showed a similar pattern of seasonal variations at all five stations, with high values in winter and low values in summer/autumn (Fig. 8(c)). Seasonal amplitude was the lowest in Miyako and the highest in Hinase and Ohno (Fig. 7(a)). In Hinase and Ohno, annual range of pH$_{min}$ variation reached over 1.1 pH units, which was larger than that of 83 stations around the world listed in Carstensen and Duarte (2019) except for Baltic Sea. This seasonal variation overlapped with the short-term drawdown event of pH$_{min}$, which was frequently observed in summer/autumn. The timing of the amplitude of such events

was synchronised with that of short-term low-salinity events (Fig. 7(b)). This result indicates that either rainfall or increased river flow causes several short-term processes in these coastal areas, causing short-term variation in pH.

### 3.3 Relationship between TA and salinity

Figure 9 shows the observed relationship between
TA and salinity based on discrete water samples obtained at each station. TA in Miyako, Kashiwazaki, and Shizugawa varied in a narrow range between 2,222 and 2,236 μmol kg$^{-1}$ at a salinity of 33.5, which is approximately equal to that in the surface waters of the western North Pacific in the
corresponding latitudinal range (Takatani et al., 2014). This similarity of TA range indicates that coastal waters in these three areas are occasionally not significantly different from their open water sources when salinity is satisfactorily high (~33.5). In contrast, in Hinase and Ohno, we obtained TA
of 2,222 and 2182 μmol kg$^{-1}$, respectively, if we extended their regression equation to a salinity of 34.0. These values again fit with those of Kuroshio waters (Takatani et al., 2014), reflecting that the ultimate source water for the Seto Inland Sea is Kuroshio. It is noteworthy that the maximum
salinity values observed in Hinase and Ohno were 31.67 and 31.43, respectively (Fig. 7(b)). These results indicate that the open waters of the Hinase and Ohno areas are already diluted from their ultimate source (i.e., Kuroshio water), and hence, have already witnessed several modifications
through coastal processes within the Seto Inland Sea.

The freshwater endmember in each regression line represents the TA of the main freshwater source in each area, although the absence of a low-salinity data point leads to a high uncertainty in estimating the endmember. Table 1

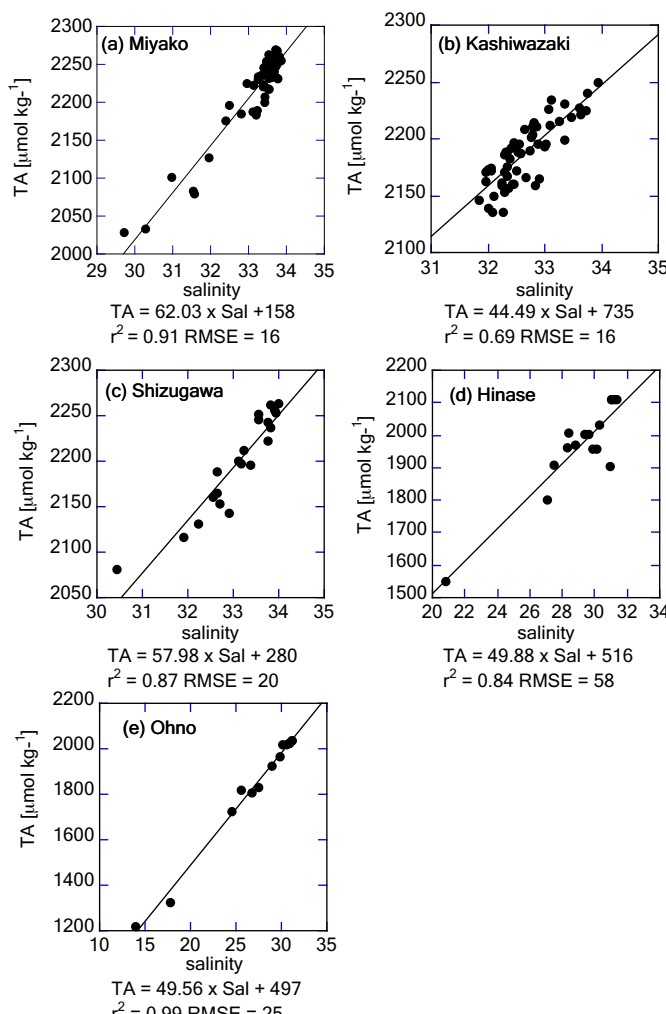

Figure 9: Salinity–TA relationship based on discrete water samples obtained from (a) Miyako, (b) Kashiwazaki, (c) Shizugawa, (d) Hinase, and (e) Ohno stations. Regression equations are provided below the X-axes.

describes the list of TA at the mouth of the major freshwater inflow in each coastal area. In most study areas, calculated freshwater endmember of TA-S regressions agreed with the observed TA of one of freshwater source (Table 1), indicating that

non-dilution/concentration changes of TA (Jiang et al. 2014) was relatively small in these areas, and by comparing these values with the freshwater endmember calculated from Fig. 9, we can speculate which river was the actual major freshwater source for each area. For the Miyako station, which is located at the northern edge of Miyako Bay, the Tashiro River becomes a major

controller of salinity because of its proximity to the station. Two small rivers are neighbouring the Kashiwazaki site, and both rivers can act as salinity controllers for this site based on TA. Interestingly, all small rivers that flow into Shizugawa Bay had far higher TA than the calculated freshwater endmember, reflecting the existence of limestone areas in their basin. Rather, the TA of Kitakami River, despite flowing into a different bay south of Shizugawa, fitted with that of the freshwater endmember, suggesting that this river is the main source of freshwater for Station S3 because of its large flow rate. In the Hinase Archipelago,

all surrounding rivers had similar TA, and hence, we cannot determine which rivers are the main freshwater sources for Station H2. The freshwater endmember calculated in Ohno Strait was significantly higher than TA of its neighbouring large river (Ohta River), suggesting an additional contribution from small rivers directly facing this strait.

Table 1. TA of freshwater sources in each coastal area, with calculated TA of freshwater endmember from the salinity–TA
relationship (Fig. 9). Uncertainty terms of freshwater endmember was calculated as standard error of sal = 0 intercept in least squares fitting.

| Area | Freshwater endmember of TA from Fig. 9 | Freshwater source in each area | TA in each freshwater source |
|---|---|---|---|
| Miyako Bay | $158\pm84$ µmol kg$^{-1}$ | Hei River | 415 µmol kg$^{-1}$ [a] |
| | | Tsugaruishi River | 482 µmol kg$^{-1}$ [b] |
| | | Tashiro River | 142 µmol kg$^{-1}$ [b] |
| Kashiwazaki Coast | $735\pm132$ µmol kg$^{-1}$ | Sabaishi River | 845 µmol kg$^{-1}$ [b] |
| | | Ukawa River | 655 µmol kg$^{-1}$ [b] |
| Shizugawa Bay | $280\pm172$ µmol kg$^{-1}$ | Kitakami River | 351 µmol kg$^{-1}$ [a] |
| | | Small rivers in Shizugawa Bay | 725–1108 µmol kg$^{-1}$ [b] |
| Hinase Archipelago | $516\pm169$ µmol kg$^{-1}$ | Chikusa River | 525 µmol kg$^{-1}$ [a] |
| | | Small rivers in the Archipelago | 495–640 µmol kg$^{-1}$ [b] |
| Ohno Strait | $497\pm36$ µmol kg$^{-1}$ | Ohta River | 291 µmol kg$^{-1}$ [a] |

a: values referred from Kobayashi (1960)
b: measured in this study

## 3.4 Temporal variation of parameters derived from pH$_{min}$ and TA

Based on the regression equation obtained from Fig. 9, we calculated the time series of TA from the time series of salinity at each station (Fig. 10(a)). We then calculated the time series of DIC, $p$CO$_2$, $\Omega_{ar}$, and $\Omega_{ca}$ from pH$_{min}$ and TA, and here, we show the results of $p$CO$_2$ and $\Omega_{ar}$ in Fig. 10(b) and Fig. 10(c). In this calculation we used time series of pHmin and TA estimated from salinity, and uncertainty of these parameters were in the order of ±0.010 pH units and ±20 µmol kg$^{-1}$ for pH$_{min}$ and TA, respectively (see Sections 3.2 and 3.3 for details). Uncertainty of derived parameters by using these values were estimated by using CO2sys program as ±15 µatm and ±0.07 for $p$CO$_2$ and $\Omega_{ar}$, respectively.

Since TA was calculated as a linear equation of salinity, it was expected to have a resemblance between the TA and salinity patterns (Fig. 10(a)). Sporadic decreases in TA corresponding to low salinity events in Hinase and Ohno were noticeable. Water with low TA has low buffer capacity, and hence, it has particularly high risks of low pH and high $p$CO$_2$ (e.g., Salisbury, 2008; Carstensen and Duarte, 2019). Figure 10(b) shows an appearance of such risk, extremely high $p$CO$_2$ over 1,500 µatm during the low salinity period in Hinase and Ohno. The annually averaged $p$CO$_2$ values were 404, 450, 393, 589, and 621 µatm for Miyako, Kashiwazaki, Shizugawa, Hinase, and Ohno, respectively, showing that surface waters are in equilibrium with or even higher than current atmospheric concentration of CO$_2$ (416 ppm at 2021; WDCGG, 2023). Generally, estuary areas tend to become sources of atmospheric CO$_2$, while "open" coastal areas such as marginal seas, continental shelves, and large bays tend to become sinks of atmospheric CO$_2$ (e.g., Borges et al., 2005; Laruelleet al., 2010; Kubo et al., 2017; Tokoro et al., 2020). Typically, terrestrial input of organic matter from rivers is mostly decomposed in estuaries, causing high $p$CO$_2$ and low pH, as well as a high nutrient flux to the outer estuary. In an "open" coastal area, nutrients transported from the estuary activate high primary production, which causes low $p$CO$_2$ and high pH in surface waters. The Seto Inland Sea itself is considered a sink of atmospheric CO$_2$ (e.g., Tokoro et al., 2020), and hence, it can be treated as an open coastal area. Our results indicate that despite the high distance from the large river mouth, both Hinase Archipelago and Ohno Strait can be classified as "estuaries," or at least as areas that receive a significant quantity of particulate organic matter from land.

The $\Omega_{ar}$ was high in summer and low in winter in Miyako, Kashiwazaki, and Shizugawa (Fig. 10(c)), showing an opposite pattern to the seasonal variation of pH$_{min}$. This was due to the high seasonal variation in the solubility of aragonite induced by the seasonal change in water temperature (see Section 4.2). In contrast, in Hinase and Ohno, the amplitude of seasonal variation of pH was high enough to overcome seasonal variation of aragonite solubility, and as a result, $\Omega_{ar}$ changed to show a seasonal maximum in winter and a seasonal minimum in summer. Short-term variation of $\Omega_{ar}$ linked to that of salinity also overlapped this seasonal scale variation, and as a result, significantly low $\Omega_{ar}$ conditions were frequently observed in Hinase and Ohno. Fujii et al. (2023) described that during summer–autumn in the Hinase Archipelago, $\Omega_{ar}$ occasionally falls below the threshold level of the larvae of the Pacific oyster *Crassostrea gigas*, with the effect of ocean acidification (OA) becoming detectable in the rearing experiment (~$\Omega_{ar}$ = 1.5; Waldbusser et al., 2015). This study showed that the situation was almost the same in Ohno Strait (Fig. 10(c)). It should be highlighted that Fujii et al. (2023) also noted that actual damaged

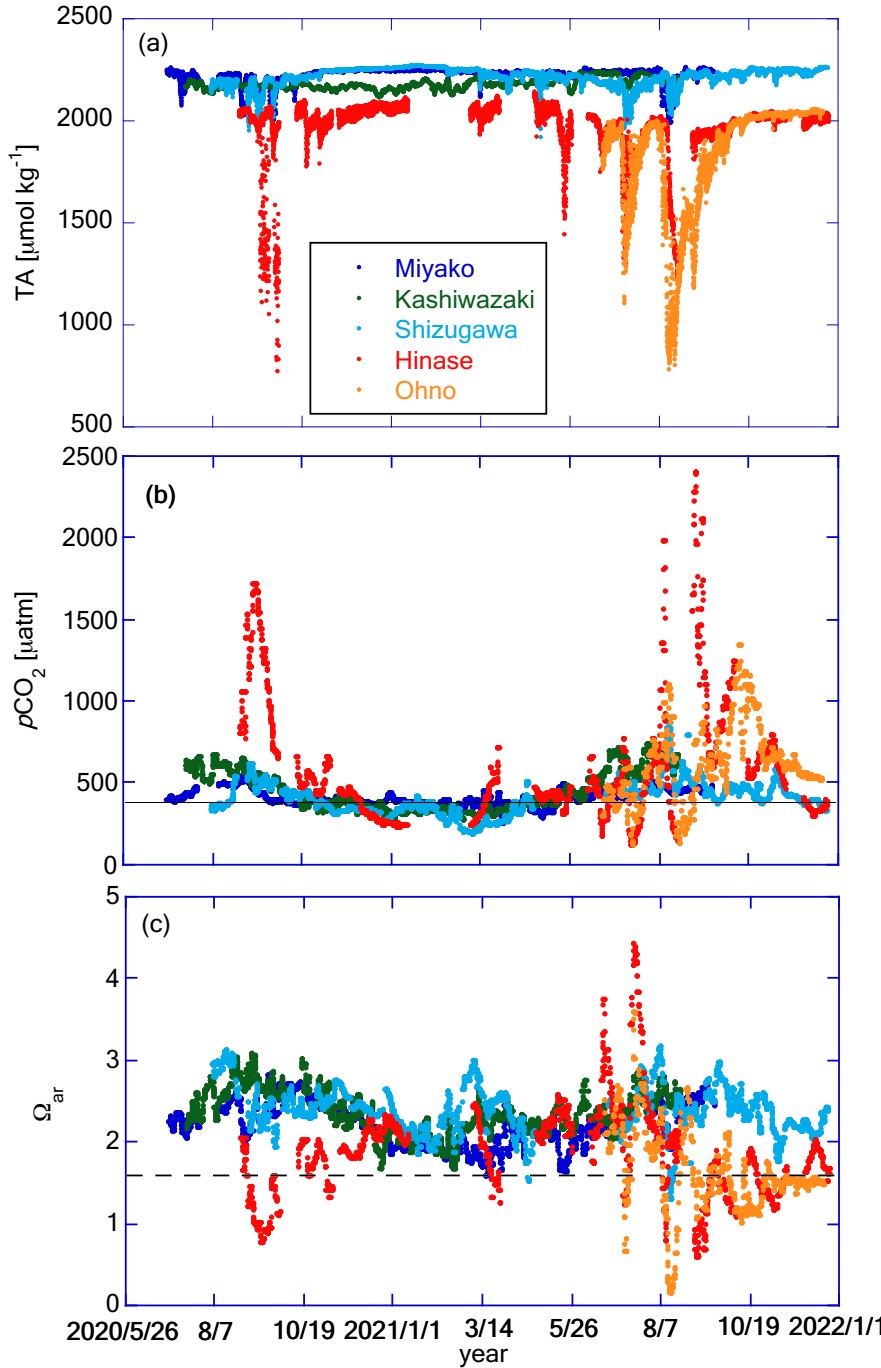

**Figure 10: Time series of (a) TA, (b) $p$CO$_2$, and (c) $\Omega_{ar}$ in the five stations. Legends of colour plots are the same for all panels as shown in Fig. 10(a). Solid line in Fig. 10(b) represents the $p$CO$_2$ value in equilibrium with the present atmospheric CO$_2$ concentration of 416 ppm. Dashed line in Fig. 10(c) represents the experimentally obtained threshold of the ocean acidification effect for the larvae of Pacific oyster *Crassostrea gigas* ($\Omega_{ar}$ = 1.5, Waldbusser et al. [2015]).**

larvae were not detected based on microscopic inspection of Hinase (n = 1062). The rearing experiment of Waldbusser et al. [2015] was conducted on the Oregon coast, where *C. gigas* is non-native. Therefore, it is not unrealistic that there is a difference in the tolerance for OA between the local population of Oregon and Seto Inland Sea, the latter being the native habitat of *C. gigas*. Kurihara et al. (2007) examined the effect of low pH in the Seto Inland Sea population of *C. gigas,* and found that the larvae of this population are affected by OA at a pH of 7.4 (National Bureau of Standards [NBS] scale). This threshold approximately corresponds to a pH of 7.27 in the total scale, and in this case, larvae of Pacific oysters are considered to be the same in most seasons both in Hinase and Ohno (Fig. 8(c)). In the experiment by Kurihara (2007), the rearing experiment was occupied only at the control (pH of 8.2 in the NBS scale) and acidified (pH of 7.4 in the NBS scale) conditions, and hence, true threshold for the Seto Inland Sea Pacific oyster population can exist between a pH of 7.27 and 8.07. We thus need further studies, including new rearing experiments, to determine why Pacific oyster larvae are still safe in the present Hinase Archipelago.

A low $\Omega_{ar}$ level below 1.6 was also detected in Shizugawa Bay (Fujii et al., 2023; see also Table 2 and Fig. 10(c) in this study), and Miyako Bay (Table 2 and Fig. 10(c)), but its duration was only four days and one day, respectively, throughout the study period. In Kashiwazaki, $\Omega_{ar}$ was above this level throughout the observation period. Other calcifiers are important for fisheries around the Japanese coast, such as Ezo abalone (*Haliotis discus hannai*) and short-spined sea urchin (*Strongylocentrotus intermedius*), have lower $\Omega_{ar}$ thresholds than that of Pacific oyster, 1.1 (Onitsuka et al., 2018) for *H. discus hannai* and 1.12 (Zhan et al., 2016) for *S. intermedius*. Therefore, Shizugawa Bay, Miyako Bay, and the Kashiwazaki Coast have a low risk of OA, at least from the viewpoint of fisheries. We, however, should note that we have investigated only a few marine species so far, and there may be many unknown species that are vulnerable to low pH/low $\Omega_{ar}$ condition. We must enhance our knowledge of biological responses according to species, especially for those with low economic importance, such that we can evaluate the total risk posed by OA to the coastal ecosystem.

## 4 Discussion

### 4.1 Quantification of short-term variability in properties of water

We focused on short-term variations of pH and related parameters with timescales shorter than one month, which cannot be detected by regular MOE monitoring based on water sampling. To assess this quantitatively, we calculated the SD of each parameter at different timescales (annual, monthly, and 10-days) as follows (see also Appendix 1):

| | |
|---|---|
| $SD_a$ | annual standard deviation of each data |
| $SD_{a<m>}$ | annual standard deviation of monthly average |
| $SD_m$ | monthly standard deviation of each data |
| $avgSD_m{}^a$ | annual average of $SD_m$ |

| 460 | maxSD$_m^a$ | annual maximum of SD$_m$ |
| | SD$_{10}$ | moving 10-day standard deviation of each data |
| | avgSD$_{10}^a$ | annual average of SD$_{10}$ |
| | maxSD$_{10}^a$ | annual maximum of SD$_{10}$ |
| | avgSD$_{10}^m$ | monthly average of SD$_{10}$ |

The results are listed in Table 2. As shown in Figures 7 and 9, both the amplitude and frequency of short-term variation differed significantly among the seasons for almost all parameters. To determine seasonal differences in the extent of short-term variation, avgSD$_{10}^m$ in each month was further listed in Table 3.

      The SD$_{a<m>}$ of water temperature were in the range of 3.9 - 7.2, and this was almost same with that of SD$_a$ (Table 2). This implies that large-time scale variation over one month (ca., seasonal scale) was the main component of observed annual
temporal variation (Fig. 7(a)). The avgSD$_m^a$ of water temperature (0.9–1.5) were < 25% of the SD$_a$ (3.9–7.2), and the avgSD$_{10}^a$ (0.5–0.8) was approximately half of avgSD$_m^a$ in each area (Table 2). In the case of salinity, on the other hand, SD$_{a<m>}$ (0.34 - 3.91) was about 70 % of SD$_a$ (0.63 - 5.34), indicating that variations with shorter time scale than one month contributes some parts of annual variation in the case of salinity. The avgSD$_m^a$ of salinity (0.33–2.37) was approximately half of the SD$_a$ (0.66–5.34) in each area, and avgSD$_{10}^a$ (0.26–1.40) was approximately two thirds of avgSD$_m^a$. Seasonal variability of avgSD$_{10}^m$ was
also higher than that of water temperature (Table 3), and as a result, avgSD$_{10}^m$ of salinity occasionally exceeded the SD$_a$ in several months.  Interestingly, avgSD$_{10}^a$ and avgSD$_m^a$ were almost the same in Miyako, Kashiwazaki, and Shizugawa (Table 2), and the maxSD$_{10}^a$ for salinity was higher than the maxSD$_m^a$. Such a result can be attributed to the perturbation of salinity that mainly occurs within the timescale of 10-days, and monthly SD of salinity is essentially determined by the amplitude and frequency of 10-days salinity variation included within that month. Although the avgSD$_{10}^a$ of salinity was lower than avgSD$_m^a$
in Hinase and Ohno, the maxSD$_{10}^a$ was still higher than the maxSD$_m^a$., indicating that the 10-days variation is also an influential component of monthly variation in these two areas.

      The relative contribution of short-term variation to annual variability for other parameters varied between the above two extremes (water temperature and salinity). In the cases of DO and pH, SD$_{a<m>}$ was on the same level with SD$_a$ in Miyako, Kashiwazaki and Shizugawa, but it declined to about 80 % of SD$_a$ in Hinase and Ohno.  The avgSD$_m^a$ was around 25 - 30%
of the SD$_a$ in Miyako and Kashiwazaki, but it increased to over 50% in Shizugawa, Hinase, and Ohno both for DO and pH. The maxSD$_m^a$ exceeded SD$_a$ in Hinase and Ohno both for DO and pH. Even the maxSD$_{10}^a$ exceeded SD$_a$ in Ohno in the case of DO, while it exceeds in Shizugawa, Hinase, and Ohno in the case of pH (Table 2). These results indicate that although the main driver of the annual variation for DO and pH were seasonal variation in Miyako and Kashiwazaki, shorter variation also contributed in Shizugawa, Hinase, and Ohno stations. In the cases of $\Omega_{ar}$, SD$_{a<m>}$ showed lower values than SD$_a$ in all stations,
indicating larger contribution of short-term variation compared to the case of DO and pH. The avgSD$_m^a$ was approximately half of SD$_a$ for all stations, while the avgSD$_{10}^a$ was approximately one third of the avgSD$_m^a$. Both maxSD$_m^a$ and maxSD$_{10}^a$ exceeded SD$_a$ in most areas. These results indicate that short-term variation in the 10-days scale is the significant driver of

Table 2. Statistical aspects of environmental parameters in each station. Annual values are calculated based on the last 1-year data to avoid biases that come from different time length.

| Area | Miyako | Kashiwa-zaki | Shizu-gawa | Hinase | Ohono |
|---|---|---|---|---|---|
| **Water temp. [°C]** | | | | | |
| $SD_a$ | 4.9 | 7.2 | 5.6 | 6.6 | 3.9 |
| $SD_{a<m>}$ | 4.8 | 7.0 | 5.6 | 6.6 | 3.9 |
| $avgSD_m{}^a$ | 0.9 | 1.2 | 1.3 | 1.5 | 1.3 |
| $maxSD_m{}^a$ | 2.1 | 2.2 | 2.2 | 2.7 | 2.1 |
| $avgSD_{10}{}^a$ | 0.5 | 0.6 | 0.6 | 0.6 | 0.8 |
| $maxSD_{10}{}^a$ | 1.5 | 1.9 | 2.3 | 1.9 | 2.1 |
| | | | | | |
| **Salinity** | | | | | |
| $SD_a$ | 0.66 | 0.63 | 0.79 | 2.44 | 5.34 |
| $SD_{a<m>}$ | 0.34 | 0.50 | 0.64 | 1.79 | 3.91 |
| $avgSD_m{}^a$ | 0.33 | 0.37 | 0.43 | 1.28 | 2.37 |
| $maxSD_m{}^a$ | **1.10** | **0.64** | **1.04** | **5.67** | **7.47** |
| $avgSD_{10}{}^a$ | 0.26 | 0.25 | 0.37 | 0.81 | 1.40 |
| $maxSD_{10}{}^a$ | **1.31** | **0.73** | **1.71** | **6.00** | **7.93** |
| | | | | | |
| **Oxygen [μmol kg$^{-1}$]** | | | | | |
| $SD_a$ | 60 | 51 | 26 | 53 | 44 |
| $SD_{a<m>}$ | 60 | 50 | 26 | 40 | 37 |
| $avgSD_m{}^a$ | 17 | 13 | 13 | 36 | 35 |
| $maxSD_m{}^a$ | 28 | 20 | 22 | **62** | **57** |
| $avgSD_{10}{}^a$ | 9 | 7 | 10 | 11 | 22 |
| $maxSD_{10}{}^a$ | 22 | 17 | 21 | 40 | **48** |
| | | | | | |
| **$pH_{min}$** | | | | | |
| $SD_a$ | 0.05 | 0.09 | 0.09 | 0.21 | 0.16 |
| $SD_{a<m>}$ | 0.04 | 0.09 | 0.08 | 0.16 | 0.11 |
| $avgSD_m{}^a$ | 0.02 | 0.03 | 0.05 | 0.10 | 0.09 |
| $maxSD_m{}^a$ | **0.06** | 0.05 | **0.09** | **0.23** | **0.24** |
| $avgSD_{10}{}^a$ | 0.02 | 0.02 | 0.03 | 0.06 | 0.07 |
| $maxSD_{10}{}^a$ | **0.06** | 0.04 | **0.10** | **0.26** | **0.31** |
| | | | | | |
| **$\Omega_{ar}$** | | | | | |
| $SD_a$ | 0.25 | 0.26 | 0.29 | 0.58 | 0.52 |
| $SD_{a<m>}$ | 0.24 | 0.23 | 0.16 | 0.41 | 0.33 |
| $avgSD_m{}^a$ | 0.12 | 0.14 | 0.23 | 0.31 | 0.31 |
| $maxSD_m{}^a$ | 0.21 | 0.22 | **0.48** | **0.89** | **0.78** |
| $avgSD_{10}{}^a$ | 0.09 | 0.11 | 0.15 | 0.22 | 0.25 |
| $maxSD_{10}{}^a$ | **0.25** | 0.21 | **0.63** | **1.27** | **0.81** |

Numbers are shown in bold when they exceed $SD_a$.

Table 3. Monthly average of 10-day standard variation ($avgSD_{10}^m$) for each parameter on each month in each station
Numbers shown by bold represent annual maximum and the second maximum

| Area | Parameter | Jan. | Feb. | Mar. | Apr. | May | Jun. | Jul. | Aug | Sept. | Oct. | Nov. | Dec |
|---|---|---|---|---|---|---|---|---|---|---|---|---|---|
| Miyako | | | | | | | | | | | | | |
| | Water Temp. [°C] | 0.26 | 0.17 | 0.15 | 0.28 | 0.64 | **0.72** | **0.86** | 0.58 | 0.70 | 0.53 | 0.35 | 0.45 |
| | Salinity | 0.03 | 0.12 | 0.28 | 0.17 | 0.15 | 0.10 | 0.32 | **0.71** | **0.78** | 0.29 | 0.08 | 0.05 |
| | oxygen [µmol kg$^{-1}$] | 8.9 | 13.1 | 10.8 | **14.4** | **13.5** | 8.4 | 9.5 | 6.0 | 7.3 | 5.2 | 3.3 | 5.6 |
| | pH | 0.01 | 0.02 | **0.03** | **0.03** | **0.03** | 0.02 | 0.01 | 0.01 | 0.01 | 0.01 | 0.01 | 0.01 |
| | Ωar | 0.03 | 0.08 | 0.11 | **0.14** | 0.12 | **0.14** | 0.08 | 0.13 | 0.13 | 0.07 | 0.04 | 0.04 |
| Kashiwazaki | | | | | | | | | | | | | |
| | Water Temp. [°C] | 0.70 | **0.76** | 0.34 | 0.34 | 0.51 | 0.55 | 0.61 | 0.57 | 0.58 | 0.74 | 0.59 | **1.08** |
| | Salinity | 0.25 | **0.41** | 0.20 | 0.20 | **0.38** | 0.15 | 0.37 | 0.20 | 0.13 | 0.21 | 0.18 | 0.30 |
| | oxygen [µmol kg$^{-1}$] | 9.9 | **11.7** | 5.0 | 4.2 | 6.1 | 6.9 | 5.3 | 3.8 | 4.1 | 6.0 | 5.8 | **10.8** |
| | pH | 0.02 | 0.02 | 0.02 | 0.02 | 0.02 | **0.03** | 0.02 | 0.02 | 0.02 | 0.02 | 0.02 | **0.03** |
| | Ωar | 0.08 | **0.13** | 0.09 | 0.08 | 0.10 | 0.08 | **0.13** | 0.12 | 0.10 | 0.12 | 0.11 | **0.15** |
| Shizugawa | | | | | | | | | | | | | |
| | Water Temp. [°C] | 0.30 | 0.32 | 0.23 | 0.61 | 0.94 | **1.15** | 0.68 | **1.13** | 0.61 | 0.46 | 0.44 | 0.40 |
| | Salinity | 0.04 | 0.13 | 0.32 | 0.58 | 0.56 | 0.32 | **0.77** | 0.61 | **0.66** | 0.23 | 0.18 | 0.09 |
| | oxygen [µmol kg$^{-1}$] | -- | -- | -- | -- | **16.8** | 10.1 | 9.0 | **13.3** | 7.4 | 7.0 | 5.6 | 9.1 |
| | pH | 0.02 | **0.04** | 0.03 | **0.04** | 0.03 | 0.02 | 0.03 | **0.05** | 0.03 | 0.02 | 0.02 | 0.01 |
| | Ωar | 0.10 | 0.18 | 0.17 | 0.18 | 0.15 | 0.14 | 0.18 | **0.27** | 0.16 | **0.20** | 0.13 | 0.06 |
| Hinase | | | | | | | | | | | | | |
| | Water Temp. [°C] | **0.88** | -- | 0.51 | 0.41 | 0.60 | 0.34 | 0.71 | **1.00** | 0.49 | 0.71 | 0.65 | 0.72 |
| | Salinity | 0.05 | -- | 0.19 | 0.20 | 0.61 | 1.33 | **2.41** | **2.17** | 1.16 | 0.40 | 0.28 | 0.10 |
| | oxygen [µmol kg$^{-1}$] | -- | -- | -- | -- | -- | -- | -- | -- | 16.4 | 13.1 | 8.2 | 7.5 |
| | pH | 0.01 | -- | 0.06 | -- | 0.02 | 0.09 | **0.12** | **0.15** | 0.08 | 0.04 | 0.03 | 0.02 |
| | Ωar | 0.07 | -- | 0.19 | -- | 0.11 | **0.38** | **0.58** | 0.34 | 0.20 | 0.14 | 0.10 | 0.09 |
| Ohno | | | | | | | | | | | | | |
| | Water Temp. [°C] | -- | -- | -- | -- | -- | **1.26** | 0.90 | **1.13** | 0.61 | 0.54 | 0.49 | 0.44 |
| | Salinity | -- | -- | -- | -- | -- | 0.68 | **2.75** | **3.60** | 2.03 | 0.44 | 0.22 | 0.10 |
| | oxygen [µmol kg$^{-1}$] | -- | -- | -- | -- | -- | 17.9 | 29.6 | **31.0** | 29.3 | **29.7** | 9.5 | 4.6 |
| | pH | -- | -- | -- | -- | -- | 0.04 | 0.09 | **0.14** | **0.10** | 0.07 | 0.02 | 0.01 |
| | Ωar | -- | -- | -- | -- | -- | 0.26 | **0.46** | **0.47** | 0.27 | 0.18 | 0.07 | 0.04 |
| Tokyo Bay* | | | | | | | | | | | | | |
| | Water Temp. [°C] | 0.36 | 0.31 | 0.55 | 0.45 | 0.56 | 0.76 | **0.84** | **0.96** | 0.82 | 0.57 | 0.28 | 0.51 |
| | Salinity | 0.34 | 0.35 | 0.79 | 0.72 | 0.67 | 1.35 | **2.11** | **1.65** | 1.41 | 0.86 | 0.53 | 0.46 |
| | pH | 0.02 | 0.04 | 0.05 | 0.07 | 0.08 | **0.15** | **0.15** | 0.13 | 0.13 | 0.08 | 0.04 | 0.02 |

*See Secion 4.3 for the detail of the data in Tokyo Bay.

annual temporal variation for $\Omega_{ar}$, which was similar to the case of salinity.

We further investigated the seasonal dependency of the 10-days SD in each area based on Table 3. The $maxSD_{10}^a$ for DO, $pH_{min}$, and $\Omega_{ar}$ occurred at approximately the same time, and moreover, seasonal distribution of $avgSD_{10}^a$ showed positive correlation with statistical significance ($r^2 > 0.5$) among these three parameters. These results indicate that the 10-days variations of these three parameters were caused by the same processes (ca. biological activities).

However, the specific timing of the occurrence of the annual maximum for these three parameters differed among the areas. In Miyako, the annual maximum of $avgSD_{10}^m$ for DO, $pH_{min}$, and $\Omega_{ar}$ occurred in spring, when biological activities are the highest. In Shizugawa, Hinase, and Ohno, the maximum $avgSD_{10}^m$ for these parameters occurred during late summer and early autumn, which approximately overlapped with the timing of the occurrence of the annual maximum of $avgSD_{10}^m$ of salinity (Table 3). In Kashiwazaki, the maximum $avgSD_{10}^m$ for DO, $pH_{min}$, and $\Omega_{ar}$ occurred in winter, when strong perturbation of river flow induced by heavy snowfall caused an annual maximum $avgSD_{10}^m$ for both salinity and water temperature. Overall, the investigated statistical aspects of short-term variations indicated that several physical processes related to the 10-days salinity variation had derivatively induced biological processes that caused short-term variations in DO, $pH_{min}$, and $\Omega_{ar}$ in these coastal areas.

Temporal variation of pH and the related parameters are mainly analysed with interannual, seasonal and diurnal time scales (e.g., Bauman and Smith 2018; Carstensen et al., 2019; Hassoun et al., 2017; Lowe et al., 2018; Provoost et al., 2010; Rosenau et al., 2021), but many studies have also highlighted existence of significant day-to-day scale pH variation in coastal areas, especially in estuaries (e.g., Frieder et al., 2012; Fujii et al., 2021; Hoffman et al., 2012; Johnson et al., 2013; Bednarzek et al., 2022). The present results illustrated that such day-to-day variation with the timescales from 10 to 30 days plays significant contribution to the annual variation of pH and the related parameters in the coastal areas in Japan. Indeed, amplitude of $SD_a$ and $avgSD_m^a$ of $pH_{min}$ observed in Hinase and Ohno was almost similar with those observed in US estuaries by former study (Hoffman et al., 2011; Bednarzek et al., 2022), indicating that such high contribution of short-term variation to total variability is common amongst the world's estuaries. In several cites off California coasts, it was reported that eventual coastal upwelling brings carbon and nutrient to the coast and cause multi-days scale pH variation (e.g., Frieder et al., 2012; Kessouri et al., 2021). In Japan coasts, on the other hand, the observed short-term variations in pH seemed to be connected strongly with that of salinity, suggesting that the change in riverine input plays significant rolls in the short-term pH variation.

**4.2 Classification of temporal variations into thermodynamic and non-thermodynamic components**

To determine the processes that contribute to the short-term variation in biogeochemical properties, we divided the observed temporal variations of each parameter into thermodynamic and non-thermodynamic components using the following equation:

$$C_i = C_i(eq) + C_i(diseq) \hspace{6cm} (1)$$

where $C_i$ represents the observed concentration of parameter i, while $C_i(eq)$ represents the estimated concentration of parameter i in equilibrium with the current atmosphere under the observed water temperature and salinity. $C_i(diseq)$ represents the difference between $C_i$ and $C_i(eq)$. This is the similar idea of AOU, while DO(diseq) is equivarent to AOU multiplied by -1. For DO, DO(eq) was calculated from water temperature and salinity based on the formulation of Weiss (1970) with a fixed

atmospheric pressure of 1 atm. Both pH(eq) and $\Omega_{ar}$(eq) were calculated using the CO2sys program with the same set of constants as described in Section 2.1, using water temperature, salinity, estimated TA, and a fixed atmospheric $CO_2$ mole fraction of 415 ppm.

Figure 11 shows the temporal variation of DO(eq), pH(eq), and $\Omega_{ar}$(eq), while Fig. 12 shows DO(diseq), $pH_{min}$(diseq), and $\Omega_{ar}$(diseq). We also calculated 1-year SD, monthly SD, and 10-days SD for these parameters, and the results are listed in

Table 4. The annual temporal variation was approximately the same between DO(eq) and DO(diseq) (Table 4), but their origin was significantly different. While seasonal variation played a dominant role in the temporal variation of DO(eq) (Fig. 11(a), Table 4), short-term variations played a significant role in that of DO(diseq) (Fig. 12(a), Table 4). In the case of pH, the temporal variation of pH(eq) resembles that of salinity (Figs. 7(b) and 11(b)), and the amplitude of pH(eq) (~0.25, Fig. 11(b)) was approximately one order lower than that of pH(diseq) (~1.2, Fig. 12(b)). This result indicated that the thermodynamic

process had a negligible contribution to the temporal variation of pH. Interestingly, the relative contributions of monthly SD and 10-days SD of pH(diseq) to the 1-year SD showed a similar structure to that of $pH_{min}$ (Table 4), suggesting that the short-term variation at the scale of 10-days was the main driver of annual temporal variation for $pH_{min}$(diseq), which is similar to the cases of $pH_{min}$ and salinity. Temporal variation of $\Omega_{ar}$(eq) had both seasonal- and short-term scale components (Table 4), and the pattern of seasonal variation resembled that of water temperature, while that of short-term variation resembled that of

salinity (Fig. 11(c)). $\Omega_{ar}$(diseq) showed a similar scale of 1-year variability to that of $\Omega_{ar}$(eq) (Fig. 11(c) and Fig. 12(c), Table 4), similar to the case of DO. The temporal pattern of $\Omega_{ar}$(diseq) was quite similar to that of $pH_{min}$(diseq), suggesting that short-term variation at the 10-days scale contributes significantly to the annual temporal variation of $\Omega_{ar}$(diseq), similar to the case of $pH_{min}$(diseq). Overall, the non-thermodynamic component contributed approximately half of the observed annual variation in DO and $\Omega_{ar}$, and in the case of pH, the non-thermodynamic component played a dominant role in the annual variation. For

all these biogeochemical parameters, the short-term variation at the 10-days scale contributed a significant percentage of the temporal variation of the non-thermodynamic component.

To determine the specific process that drives the non-thermodynamic temporal variation of these properties, we examined the correlation between $pH_{min}$(diseq) and DO(diseq) (Fig. 13(a)). These two properties showed significant positive correlation, indicating that the non-thermodynamic component of temporal variations was driven mainly by biological

processes. Several studies have also detected a positive relationship between variations in oxygen saturation and the metabolic component of pH in coastal waters, suggesting the primary influence of biological processes on pH dynamics (e.g., Baumann

Table 4. Statistical features of thermodynamic and non-thermodynamic components

| Area | Miyako | Kashiwa-zaki | Shizu-gawa | Hinase | Ohono |
|---|---|---|---|---|---|
| DO(eq) [$\mu$mol kg$^{-1}$] | | | | | |
| $SD_a$ | 33 | 40 | 27 | 35 | 20 |
| $SD_{a<m>}$ | 33 | 40 | 27 | 35 | 20 |
| $avgSD_m{}^a$ | 6 | 7 | 13 | 8 | 6 |
| $maxSD_m{}^a$ | 13 | 19 | 24 | 14 | 9 |
| $avgSD_{10}{}^a$ | 3 | 4 | 5 | 4 | 4 |
| $maxSD_{10}{}^a$ | 10 | 8 | 7 | 5 | 6 |
| DO(diseq) [$\mu$mol kg$^{-1}$] | | | | | |
| $SD_a$ | 25 | 14 | 25 | 25 | 40 |
| $SD_{a<m>}$ | 24 | 11 | 18 | 21 | 25 |
| $avgSD_m{}^a$ | 12 | 8 | 13 | 15 | 27 |
| $maxSD_m{}^a$ | 22 | 17 | 21 | 28 | 48 |
| $avgSD_{10}{}^a$ | 8 | 4 | 11 | 11 | 23 |
| $maxSD_{10}{}^a$ | 20 | 6 | 18 | 17 | 34 |
| $pH_{min}$(eq) | | | | | |
| $SD_a$ | 0.005 | 0.004 | 0.005 | 0.014 | 0.037 |
| $SD_{a<m>}$ | 0.003 | 0.003 | 0.004 | 0.010 | 0.026 |
| $avgSD_m{}^a$ | 0.003 | 0.002 | 0.003 | 0.008 | 0.016 |
| $maxSD_m{}^a$ | 0.008 | 0.004 | 0.007 | 0.034 | 0.057 |
| $avgSD_{10}{}^a$ | 0.002 | 0.001 | 0.002 | 0.004 | 0.009 |
| $maxSD_{10}{}^a$ | 0.010 | 0.003 | 0.004 | 0.012 | 0.029 |
| $pH_{min}$(diseq) | | | | | |
| $SD_a$ | 0.051 | 0.093 | 0.114 | 0.210 | 0.158 |
| $SD_{a<m>}$ | 0.035 | 0.093 | 0.082 | 0.153 | 0.107 |
| $avgSD_m{}^a$ | 0.024 | 0.031 | 0.049 | 0.103 | 0.090 |
| $maxSD_m{}^a$ | 0.059 | 0.050 | 0.092 | 0.253 | 0.225 |
| $avgSD_{10}{}^a$ | 0.016 | 0.021 | 0.029 | 0.063 | 0.067 |
| $maxSD_{10}{}^a$ | 0.103 | 0.028 | 0.048 | 0.156 | 0.124 |
| $\Omega_{ar}$(eq) | | | | | |
| $SD_a$ | 0.4 | 0.6 | 0.4 | 0.5 | 0.5 |
| $SD_{a<m>}$ | 0.3 | 0.6 | 0.4 | 0.4 | 0.2 |
| $avgSD_m{}^a$ | 0.1 | 0.1 | 0.1 | 0.2 | 0.3 |
| $maxSD_m{}^a$ | 0.2 | 0.2 | 0.3 | 0.8 | 0.9 |
| $avgSD_{10}{}^a$ | 0.1 | 0.1 | 0.1 | 0.1 | 0.2 |
| $maxSD_{10}{}^a$ | 0.2 | 0.1 | 0.1 | 0.3 | 0.4 |
| $\Omega_{ar}$(diseq) | | | | | |
| $SD_a$ | 0.2 | 0.4 | 0.4 | 0.8 | 0.5 |
| $SD_{a<m>}$ | 0.1 | 0.4 | 0.3 | 0.6 | 0.4 |
| $avgSD_m{}^a$ | 0.1 | 0.1 | 0.2 | 0.4 | 0.3 |
| $maxSD_m{}^a$ | 0.2 | 0.2 | 0.4 | 0.9 | 0.7 |
| $avgSD_{10}{}^a$ | 0.1 | 0.1 | 0.1 | 0.2 | 0.2 |
| $maxSD_{10}{}^a$ | 0.2 | 0.1 | 0.2 | 0.6 | 0.4 |

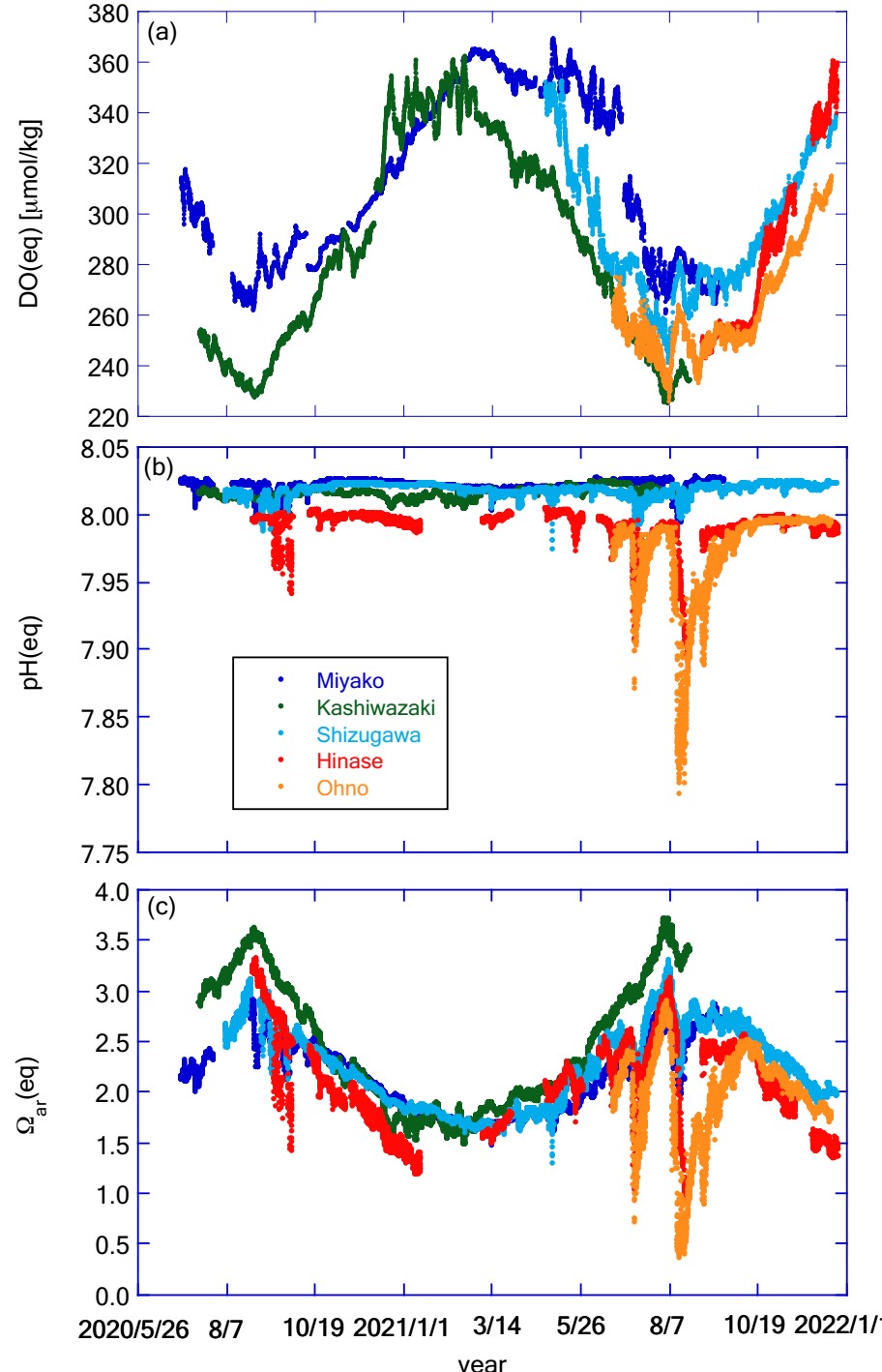


**Figure 11: Time series of (a) DO(eq), (b) pH(eq), and (c) Ω_{ar}(eq) in the five stations. Legends of colour plots are the same for all panels as shown in Fig. 11(b).**

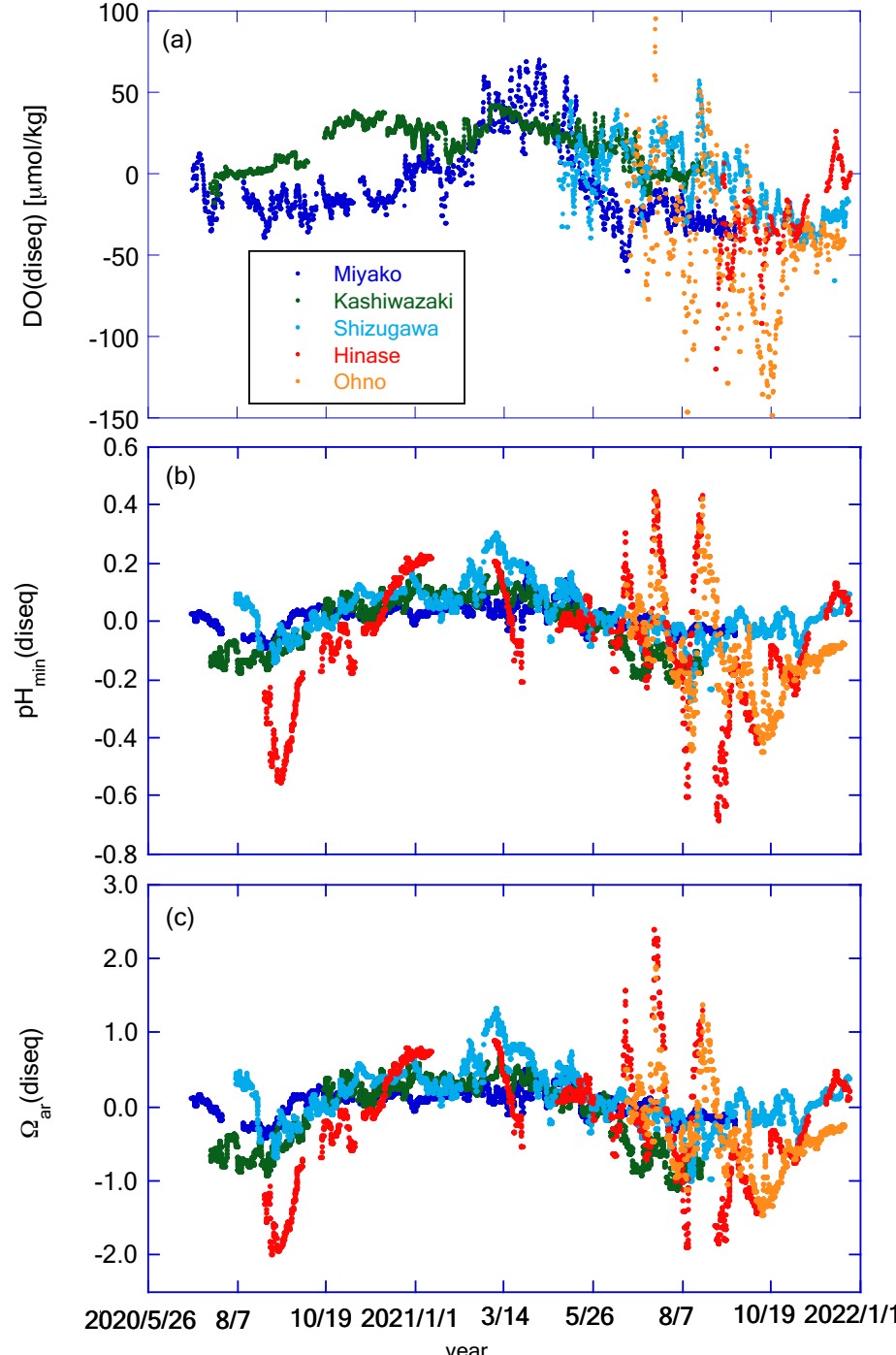

**Figure 12: Time series of (a) DO(diseq), (b) pH_min(diseq), and (c) Ω_ar(diseq) in the five stations. Legends of colour plots are the same for all panels as shown in Fig. 12(a).**


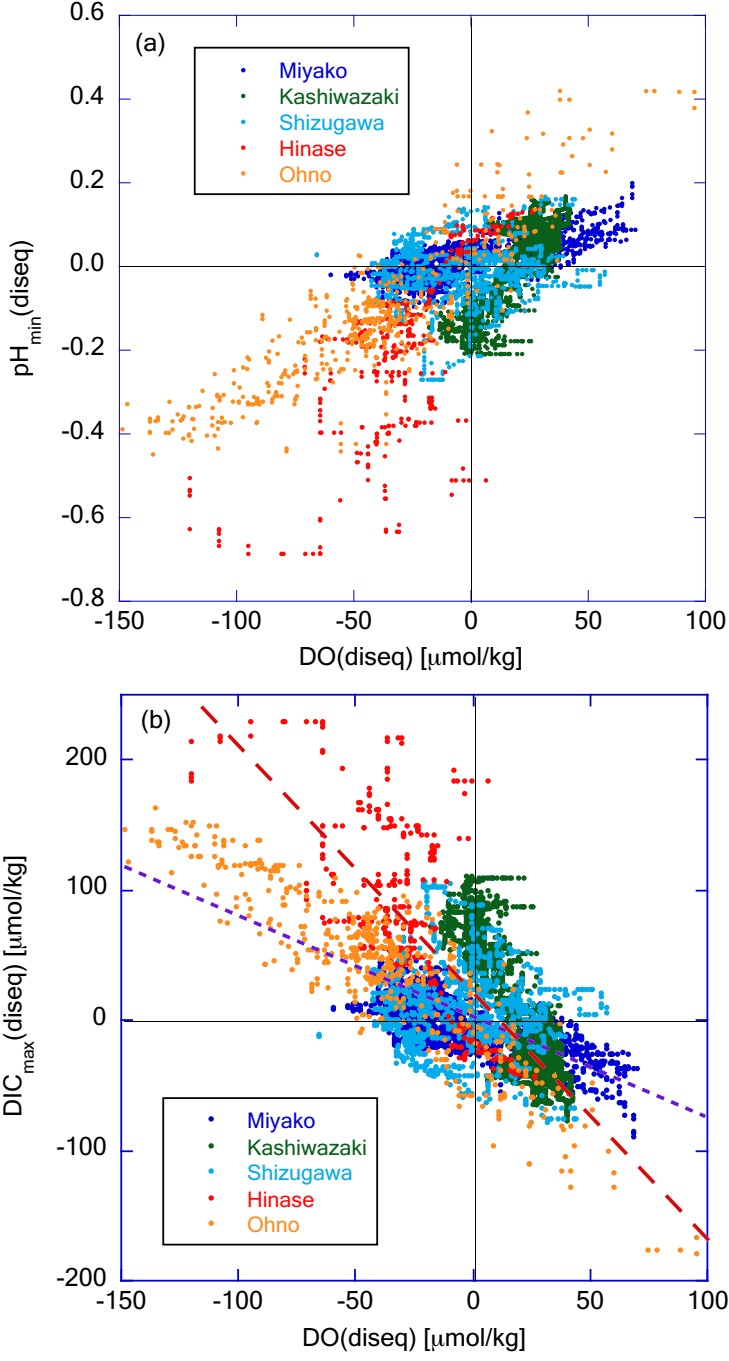

**Figure 13: Plot of (a) pH<sub>min</sub>(diseq) and (b) DIC<sub>max</sub>(diseq) against DO(diseq), respectively. The purple dashed line in Fig. 13(b) represents the theoretical line when assuming that biological production / degradation of organic matter occur with the Redfield relationship ($-\Delta$ pH<sub>min</sub>(diseq) / $\Delta$ DO(diseq) = 106/138). The red dashed line represents the regression line of Hinase data.**

and Smith, 2018; Lowe et al., 2019). We further investigated the correlation between $DIC_{max}(diseq)$ and $DO(diseq)$ (Fig. 13(b)); the former was calculated as $pH_{min}(diseq)$ and $\Omega_{ar}(diseq)$. Most data obtained in Miyako and Ohno and approximately 50% ofdata obtained in Shizugawa approximately followed a linear regression line with the slope of $-\Delta DIC_{max}(diseq)/\Delta DO(diseq)$ = 0.77 (~106/138) crossing the origin, suggesting that the observed temporal variations of both $DIC_{max}(diseq)$ and $DO(diseq)$ were induced by the production/decomposition of planktonic oceanic particles with $C : O_2$ ratio of 106 : 138 (Redfield et al.,

1963) in these two areas. Such Redfield-type relationship between short-term variation of DO and that of DIC had also been observed in the coastal area of East China Sea (Guo et al., 2021). The other data obtained in Shizugawa, as well as most data obtained in Hinase and Kashiwazaki followed the regression lines with steeper slopes than that of open ocean stoichiometry. The respiratory quotients of estuarine ecosystems can occasionally be as high as 1.5 (Wang et al., 2018) or even 2.0 (Giblin et al., 1997), when affected by the anaerobic respiration process in estuarine sediments. The observed high $-$

$\Delta DIC_{max}(diseq)/\Delta DO(diseq)$ ratio thus suggests that lateral propagation of biological processes in theneighbouring estuaries contributes significantly to the observed temporal variations of $DIC_{max}(diseq)$ and $DO(diseq)$ in these three areas. On the other hand, turnover time of oxygen in ocean mixed layer due to air-sea gas exchange (in the range from several days to two weeks; Izett et al., 2018; Qin et al., 2021) is about ten-times shorter than that of $CO_2$ (in the order of several months; e.g., Jones et al., 2014), reflecting their abundance in sea water. The $\Delta DO(diseq)$ that had once caused by biological processes, therefore, can

recover to zero with shorter time scale  than that of $\Delta DIC_{max}(diseq)$ through the air-sea gas exchange process. The observed difference in $-\Delta DIC_{max}(diseq)/\Delta DO(diseq)$ ratio between Hinase and Ohno can alco be explained by this mechanism, if large gas exchange process (ca., high wind speed and/or high wave hight) have occurred in Hinase while it had not occurred in Ohno. However, monthly-averaged wind speed was always higher in Ohno than in Hinase in 2020 autumn (Japan Meteorological Agency weather record data base, https://www.data.jma.go.jp/stats/etrn/index.php), when low $\Delta DO(diseq)$ and high

$\Delta DIC_{max}(diseq)$ were mainly observed in this study (Figure 12). Similarly, there is no significant difference in wind speed between two groups of Shizugawa data (one following Hinase data and the another following Ohno data). We thus conclude that relatively faster recovery of $\Delta DO(diseq)$ compared to that of $\Delta DIC_{max}(diseq)$ through air-sea gas exchange is not main cause of the observed high $-\Delta DIC_{max}(diseq)/\Delta DO(diseq)$ ratio, although this process may partly contribute to create high ratio in these areas.


### 4.3 Controlling factor of the amplitude of short-term pH variation.

Our analyses in Sections 3.2 and 3.4 clearly shows that severe low pH/low $\Omega_{ar}$ situations occur only at the short-term pH drawdown events that coincide with rainfall events in the coastal areas of Japan (Fig. 9(c) and 10(c)). As the amplitude of pH variation related to rainfall events differed among the five coastal areas, it is important to understand the controlling factor that

determines the amplitude of short-term pH variation. To investigate this aspect, we analysed the relationship of statistical short-

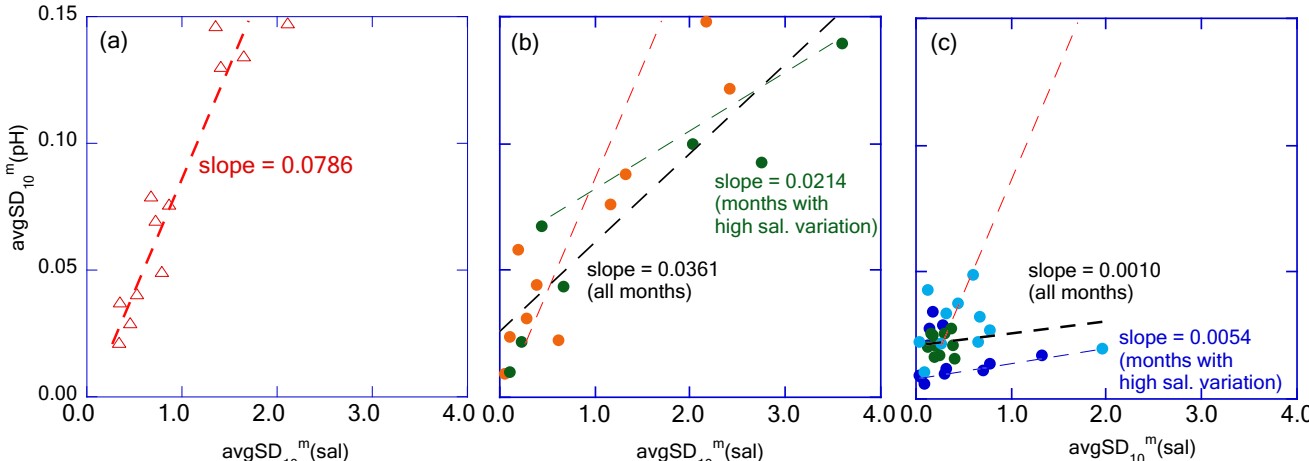

**Figure 14: Plots avgSD$_{10}^m$(pH) against avgSD$_{10}^m$(sal) for (a) Tokyo Bay, (b) Hinase and Ohno, and (c) Miyako, Kashiwazaki, and Shizugawa, respectively. The red dashed line represents regression line of Tokyo Bay, while the black dashed lines represent that of (b) Hinase and Ohno, and (c) Miyako, Kashiwazaki, and Shizugawa, respectively. Green and blue dushed lines represent regression**
**lines of Hinase and Ohno, and Miyako, Kashiwazaki, and Shizugawa, respectively, with the data of months with the avgSD$_{10}^m$(sal) higher than 1.0.**

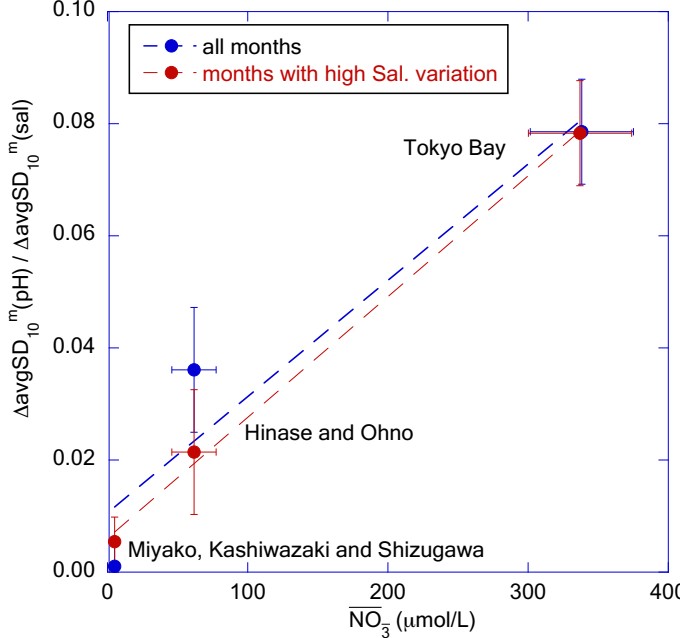

**Figure 15: Plot of ΔavgSD$_{10}^m$(pH) / ΔavgSD$_{10}^m$(sal) against $\overline{NO3}$ for each coastal area category. The blue and red dots represent**
**slope calculated from all months from the months with avgSD$_{10}^m$(sal) higher than 1.0, respectively. The dashed lines represent the regression lines.**

term variabilities between salinity and pH$_{min}$. Since short-term variation mainly occurred in the timescale of < 10 days for both salinity and pH$_{min}$ (see discussion in Section 4.1), we plotted the avgSD$_{10}^m$ of pH$_{min}$ against that of salinity (Fig. 14). In this analysis, we introduced continuous monitoring data of pH and salinity obtained in Tokyo Bay (off Kawasaki Artificial Island, https://www.tbeic.go.jp/MonitoringPost/manual/aboutObservedPoint02.pdf) to obtain information on highly eutrophic coastal areas. The details of the observation methods and the location settings of this station are described in the Appendix 2.

In Tokyo Bay, avgSD$_{10}^m$ of pH$_{min}$ (hereafter avgSD$_{10}^m$(pH)) linearly increased with an increase in avgSD$_{10}^m$ of salinity (avgSD$_{10}^m$(sal), Fig. 14(a)), suggesting that freshwater transports the sources of biological processes (both organic carbon and dissolved nutrients) at a constant concentration regardless of the flow rate. On the other hand, in Hinase and Ohno, avgSD$_{10}^m$(pH) increased with that of avgSD$_{10}^m$(sal) at the same rate as observed in Tokyo Bay when avgSD$_{10}^m$(sal) was low but $\Delta$avgSD$_{10}^m$(pH) / $\Delta$avgSD$_{10}^m$(sal) changed to a low value when avgSD$_{10}^m$(sal) exceeded 1.0 (Fig. 14(b)). Generally, suspended particle export of the river increases linearly with that of river discharge, while percentage of particle organic materials (POM) within suspended particles decreases along with the increase of suspended particle export (e.g., Coynel et al., 2005; Point et al., 2007; Zhang et al., 2013; Bukaveckas 2022). As this result, increasing rate of the POM transport with that of river discharge tends to become weak when river discharge becomes high enough (Coynel et al., 2005; Kim et al., 2020). If we assume that short term pH variation is mainly caused by the degradation of POM discharged from rivers into coastal areas and subsequential biological production by using nutrients released from the degraded POM (e.g., Salisbury et al., 2008; Kubo et al., 2017; Tokoro et al., 2020), we can expect that $\Delta$avgSD$_{10}^m$(pH) / $\Delta$avgSD$_{10}^m$(sal) will also become small at the months with high river discharge (i.e., high avgSD$_{10}^m$(sal)). The observed result in this study also indicated that the concentration of biologically active materials transported by freshwater into the Seto Inland Sea was diluted at a high freshwater flow rate, while it was not diluted in the rivers flowing into Tokyo Bay, as the latter receives far higher anthropogenic nutrients' loadings from its drainage basin than those received by the Seto Inland Sea. In the coastal area receiving further low anthropogenic nutrients' loadings such as Miyako, Kashiwazaki, and Shizugawa, $\Delta$avgSD$_{10}^m$(pH) / $\Delta$avgSD$_{10}^m$(sal) was much lower than that of Hinase and Ohno when $\overline{SD}^m_{10}$(sal) exceeded 1.0 (Fig. 14(c)).

We determined the nutrients' concentrations of the main freshwater sources for each of the three coastal area categories (i.e., Tashiro, Sabaishi, and Kitakami Rivers for Miyako + Kashiwazaki + Shizugawa, Chikusa and Ohta Rivers for Hinase + Ohno, Ara, Tama, and Tsurumi Rivers for Tokyo Bay, respectively) from the MOE Public Water Quality Database, and calculated the weighted mean nitrate concentration ($\overline{NO3}$) of river waters that flow into each coastal area using water transport as the weight. We then found that $\Delta$avgSD$_{10}^m$(pH) / $\Delta$avgSD$_{10}^m$(sal) observed at high freshwater input (that is, avgSD$_{10}^m$(sal) > 1.0) showed a linear relationship with $\overline{NO3}$ in each coastal area category (Fig. 15). Even if we compare $\Delta$avgSD$_{10}^m$(pH) / $\Delta$avgSD$_{10}^m$(sal) for all months, positive correlation against ($\overline{NO3}$) was still significant. These results imply that the amplitude of short-term variation of pH$_{min}$ in coastal areas is principally determined by the quantity of nutrients transported by freshwater from the hinterland. We, however, need to note that the riverine nitrate concentration can be an implicit function of other controlling factors. For example, we recently observed that the quantity of suspended organic

particles in river waters, as well as accumulated organic materials in riverine and estuarine sediments, are linearly correlated with the riverine nitrate concentration (unpublished data). Hence, the $\overline{NO3}$ in Fig. 15 may actually be an indicator of the transport of these materials. We thus need further detailed observations, especially at the time of increase in water level of the river, to understand practical processes that cause short-term pH variation in coastal waters.


**5 Conclusion**

In this study, we synthesised data from continuous pH monitoring of five coastal areas in Japan from 2020 to 2021. Annual variability (~1 SD) of pH and $\Omega_{ar}$ were 0.05–0.09 pH units and 0.25–0.29, respectively, for three areas with low anthropogenic nutrients' loadings (Miyako Bay, Kashiwazaki Coast, and Shizugawa Bay), while they increased to 0.16–0.21
pH units and 0.52–0.58, respectively, in two areas with medium anthropogenic nutrients' loadings (Hinase Archipelago and Ohno Strait in Seto Inland Sea). Statistical assessment of temporal variability at various timescales revealed that most of the annual variabilities in both pH and $\Omega_{ar}$ were controlled by short-term variation with a timescale of < 10 days, rather than the seasonal-scale variation. Our analyses further illustrated that most of the short-term pH variation (and hence, annual variation) was caused by biological processes, while both thermodynamic and biological processes equally contributed to the temporal
variation in $\Omega_{ar}$. Biological alteration of pH mainly occurred in oceanic areas in Miyako Bay and Ohno Strait, whereas it occurred mainly in estuarine areas in Kashiwazaki Coast and Hinase Archipelago. In Shizugawa Bay, both oceanic and estuarine areas were responsible for the biological alteration of pH.

The observed results show that short-term acidification with $\Omega_{ar}$ of < 1.5 occurred occasionally in Miyako and Shizugawa Bays, while it occurred frequently in the Hinase Archipelago and Ohno Strait. Many such short-term acidification
events were related to short-term low-salinity events. Our analyses showed that the amplitude of short-term pH variation was linearly correlated with that of short-term salinity variation, and its regression coefficient at the time of high freshwater input was positively correlated with the nutrient concentration of the main river that flows into the coastal area.

Fortunately, no marine organism that has been damaged by ocean acidification has been detected in coastal areas of Japan (Fujii et al., 2023). However, our study showed that $\Omega_{ar}$ in Japanese coastal areas occasionally drops to a level that is
potentially hazardous for marine organisms, such as Pacific oysters (< 1.5, Waldbusser et al., 2015) even in the present state. It is already known that the pH in Japanese coastal areas is decreasing at the same rate as that of the open ocean (Ishizu et al., 2019 and Ishida et al., 2021), and hence, the extent, duration, and frequency of such short-term low pH and $\Omega_{ar}$ situations will increase in coastal areas in Japan in the future (Fujii et al., 2023). Our study indicates that the amplitude of the short-term drawdown of pH related to low-salinity events will decrease if we can reduce the nutrient concentration of rivers. Similar effect
of anthropogenic nutrient reduction to diminish short-term $\Omega_{ar}$ drawdown in coastal waters are also reported by Kessouri et al. (2021). We should note, however, that a certain percentage of Japanese coastal areas are now suffering due to the problem of

low biological productivity derived from the decreased anthropogenic nutrient input (e.g., Yamamoto et al., 2021). We must consider the balance between the risk of ocean acidification and oligotrophication when we control anthropogenic loadings to coastal waters in the future. We should also note that not only the concentration of dissolved nutrients, but also other biologically active materials such as suspended organic materials in river waters and accumulated organic materials in riverine and estuarine sediments may be responsible for short-term pH variations at low salinity events (e.g., Salisbury et al., 2008; Carstensen and Duarte, 2019). Our analysis revealed that seawaters in both the Hinase Archipelago and Ohno Strait were oversaturated with $CO_2$ all through the years, and hence, it is considered that more organic matter than that biologically produced within these areas was decomposed. In such cases, not only the reduction of dissolved nutrients but also the reduction of particulate organic matter transported from rivers to coastal areas will contribute to the suppression of short-term pH drawdown. Many studies have already mentioned the contribution of seaweed/seagrass beds to the effective capture of suspended organic sediments in estuaries (e.g., Potouroglou et al., 2017; Barcelona et al., 2021; Levia-Duenas et al., 2023), and hence, the conservation and/or development of seaweed/seagrass beds in estuarine and coastal areas will contribute to the reduction of organic matter transport from rivers to coastal areas at times of high river flow. To specify effective measures against the current coastal acidification in Japan, further detailed observations, especially at the time of increase in water level of the river, are needed such that we can obtain a detailed understanding of practical processes that cause short-term pH variation in coastal waters.

## 6 Appendix 1: Schematic explanation of $SD_a$, $SD_{a<m>}$, $SD_m$, and $SD_{10}$

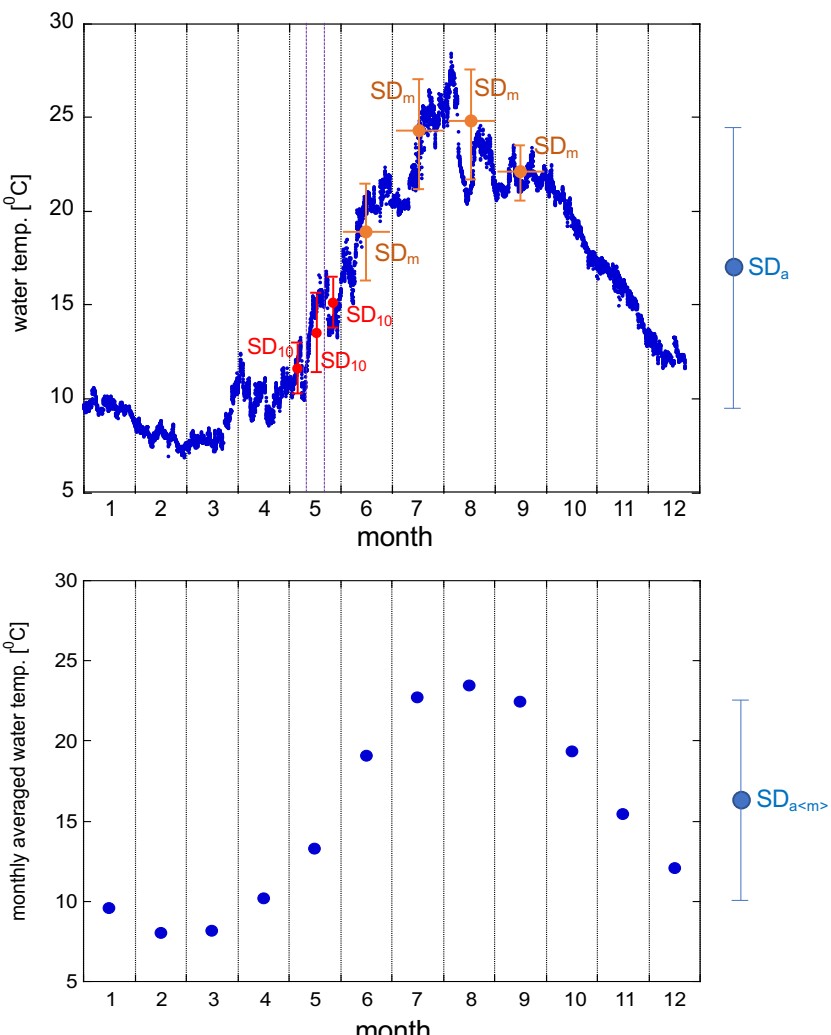

**7 Appendix 2: Observation methods and location settings of Tokyo Bay data**

Detailed data for Tokyo Bay (Kawasaki Artificial Island, Section 4.3), referred to by us, are described in the home page of data holders: https://www.tbeic.go.jp/MonitoringPost/manual/aboutObservedPoint02.pdf. Here, we provide a brief summary.

Kawasaki Artificial Island (KAI) is located in the centre of Tokyo Bay, the bay with the highest population in its basin area (2.9 million) and highest nutrient loadings (210 tN d$^{-1}$) in Japan (Fig. A1). KAI is a huge concrete cylinder with a diameter of 200 m, in which an air vending system for the trans-bay freeway tunnel is stored. An autonomous ocean profiling system

was set off KAI, which included a YSI 6600V2-4M multiple ocean sensor for the measurement of water temperature, salinity, and pH. The mechanical measurement resolution of pH was ±0.01 pH units. The sensors were cleaned and calibrated on a monthly basis. Although the pH sensor was calibrated against NBS buffers, we used pH data without conversion to a total scale, as we used these pH data only for the analysis of temporal variability. Vertical profiles were measured hourly, and the obtained data were distributed from the public database managed by the Tokyo Bay Environmental Information Center (https://www.tbeic.go.jp/MonitoringPost/ViewGraph/ViewGraph?buoyId=02). We downloaded KAI data pertaining to July 2020 to December 2021 from this database and used them for analysis.

## 8 Data Availability

Data from Tokyo Bay can be downloaded from the Tokyo Bay Environmental Information Center Public Database (https://www.tbeic.go.jp/MonitoringPost/ViewGraph/ViewGraph?buoyId=02). Other data used in this study are provided in the Supplementary Materials.

## 9 Author contribution

TO and DM performed field measurements and water sampling in Miyako Bay. MH and MY performed field measurements and water sampling on the Kashiwazaki Coast. AD performed field measurements and water sampling in Shizugawa Bay. SO, TT, MF, and RH performed field measurements and water samplings in the Hinase Archipelago. TO, GO, and KK performed field measurements and water sampling in Ohno Strait. TO undertook the measurements of discrete carbonate samples taken in Miyako Bay and Kashiwazaki Coast, while MW undertook the measurements of those taken in Shizugawa Bay, Hinase Archipelago, and Ohno Strait. Analyses of data obtained by these efforts were performed by TO.

## 10 Acknowledgements

We thank all fishers who helped us with their boats and offered their support for monitoring in Shizugawa Bay and the Hinase Archipelago. We also thank Dr. Yukihiro Nojiri, Dr. Hideaki Nakata, Dr. Osamu Matsuda, Dr. Tetsuo Yanagi, and the reviewers of this paper for their fruitful advice and discussions during our analyses. This study was funded both by the Ocean Acidification Adaptation Project (OAAP) of the Nippon Foundation and the Study of Biological Effects of Acidification and Hypoxia (BEACH) of the Environment Research and Technology Development Fund Grant Number JPMEERF20202007 of the Environmental Restoration and Conservation Agency of Japan.

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



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
