# Peer review of "Short-term variation of pH in seawaters around coastal areas of Japan: Characteristics and forcings"

_Biogeosciences, 2023_

## Author Comment (AC1)

Responses to comments from Reviewer 1

Thank you for thorough and constructive comments. Based on the comments from you and another reviewer, we have revised the manuscript as follows:

>P2, L63-65: You are mentioning that "The observed range of pH trend within the Japanese
>coast corresponds to 85% of that observed in 83 coastal systems in the world (Carstensen
>and Duarte, 2019), and this result suggests that the Japanese coastal area can function as
>the "sample shelf" of the coastal environment for the entire world." I guess you mean that
>"The observed range of pH trend within the Japanese coast is in harmony with 85% of the
>ones observed in 83 coastal systems around the world (Carstensen and Duarte, 2019)".
>Right? But this does not mean that the Japanese coastal area can represent THE word's
>coastal environment. However, it can be considered as one of the coastal areas that can be
>used as a "sample shelf" or "coastal ocean acidification test area".
In this sentence we meant to point out that the observed range of pH trend within the
Japanese coast is equivalent to 85% of that observed in 83 coastal systems in the world. Also,
I agree with your comment that Japanese coasts will be useful as a sample shelf of coastal
acidification but not that of coastal environment itself. I modified the words in this paragraph
to reflect your comments (lines 62-64).

>In P2, L60-63, you have mentioned that "The Japan Ministry of the Environment (MOE)
>conducts regular pH monitoring at over 2,000 coastal stations around Japan from the early
>1980s until the present, and the obtained data showed significant variability in the multi-
>decadal pH trend from $-0.012\,y-1$ to $+0.009\,y-1$ among the stations" and in P3, L76-81,
>you mentioned the 5 stations without justifying the reason you chose them among 2000
>stations! It would be very useful to do so before describing each station in details.
Monitoring cites in this study was not selected from the MOE stations but newly launched by
two recent pH monitoring programs (Study of Biological Effects of Acidification and Hypoxia
(BEACH) and Ocean Acidification Adaptation Project (OAAP)). The reason for
determination of monitoring locations differed by program: in BEACH, two stations were set
to represent "natural" state of Japanese coastal environment with relatively low anthropogenic
nutrient loadings. In OAAP, three stations were selected from major farming areas of pacific
oyster. We add these information in the revised text (lines 79-82).

>Also, the time range of data is different between these 5 stations. Why didn't you choose the
>ones that have exactly the same time range of beginning and end of measurements to

>facilitate the inter-comparison?

As these stations were launched in order by two different programs, we couldn't start/end years of these stations. Please extenuate that this is the first synthesis effort of OA monitoring stations operated by different founders / programs.

>P4, L111: You've mentioned that "All sensors were replaced every 2 months". Despite
>calibration? Why?

Several sensors including pH has only three-months lifetime of their batteries, and hence we changed all sensors every two months for safety. In the revised manuscript we add this information (Line 116).

>P4: You didn't mention the precision and accuracy of measurements for any parameter.
>Why? This is crucial to add.

Precision of each parameter is added in the revised text. (Lines 108-110).

>P4: You used CO2sys Excel v2.1 (Lewis and Wallace, 1998) but the correct reference should
>be (Pierrot and Wallace, 2006): Pierrot, D.E. Lewis, and D.W.R. Wallace. 2006. MS Excel
>Program Developed for CO2 System Calculations. ORNL/CDIAC-105a. Carbon Dioxide
>Information Analysis Center, Oak Ridge National Laboratory, U.S. Department of Energy,
>Oak Ridge, Tennessee. doi: 10.3334/CDIAC/otg.CO2SYS_XLS_CDIAC105a ; Index of
>/ftp/co2sys/CO2SYS_calc_XLS_v2.1 (lbl.gov).

Thank you for your comment. I modified the reference.

>P4: Also, you didn't mention the uncertainty of your carbonate system measurements. This
>can be easily calculated through the same CO2sys Excel with the new features added by Orr
>et al. (2018).

Measurement precision of DIC and Talk reported from JAMSTEC was added in Line 129.

Uncertainty of carbon-derived parameters ($pCO_2$ and $\Omega_{ara}$) are function of T, S, and carbon parameters (pH and Talk in this case), and estimated uncertainty of these parameters calculated by CO2sys were added in Section 3.4 (Lines391-400).

>You use the term "eutrophicated" in the paper. I would definitely suggest to use the
>conventional terms "eutrophic, oligotrophic, etc.".

We changed the term "eutrophicated" to "eutrophic".

>Referring to total alkalinity as Talk in the paper is very confusing. Better to type in its

>conventional abbreviation "TA".

Talk is a traditional expression of total alkalinity in the field of ocean carbon monitoring, and current major programs such as GO-SHIP still use this expression. However, I agree that TA is more used in the field of ocean acidification study, and I changed abbreviation of this parameter to TA in the revised manuscript.

>Sections 3.3. and 3.4.: You have to mention the r2 and the significance of each
>equation/relationship in each station. It would be good to add them in a separate table of in
>the plots (Figure 9).

The $r^2$ value and RMSE were added in each equation listed in Figure 9.

>Figure 7: You are showing the years in a decimal way which might be confusing for readers.
>I would suggest to add the months and the year abbreviation, e.g., Jan-20, Jan-21, etc.
>

All x-axis in Figures 7, 8, 10, 11, and 12 were changed to show date/month/year instead of years in a decimal.

>In general, the discussion needs more references, particularly for the statements used to
>explain the results, the methodology of using SDs and plotting those against each other. Also,
>this section still needs more comparison with other studies from the area and beyond.

We added several references to compare our results with former studies; such as

*annual pH ranges (Lines 318-320)

*low buffer effect in low-salinity water (Lines 403-404)

*DIC vs DO relationship (Lines 649-650)

*effect of reduction of nutrient loadings to diminish short-term drawdown of $\Omega_{ar}$ (Lines 730-732)

>P1, L20: The forcings mentioned here are already changing due to climate change, and not
> "can change". Please edit accordingly.

We changed the words to "...forcings are changing..."

>P1, L25,27: What do you mean by "anthropogenic loadings"? Could be replaced by
>"anthropogenic pressures" that encompass all types of atmospheric and terrestrial
>discharges and emissions generated by anthropogenic activities and infrastructures.

Here we meant "anthropogenic nutrient loadings," but "anthropogenic pressures" are better. In the revised text we changed these words accordingly.

>P2, L38: The sentence is not correct, as the ocean is not lowering its pH per say. Please edit
>this sentence "The ocean is lowering its pH because of anthropogenic CO2 input..." to "The
>ocean is witnessing a reduction in its pH due to anthropogenic CO2 sequestration..."
Sentence corrected.

>P2, L40: Please remove "also" after pH and edit as follows: the pH shows...
corrected.

>P2, L42: It's not only coastal upwelling but also the downwelling that is taking atmospheric
>CO2 from surface layers to the deep ocean. I would suggest to replace it with "coastal
>dynamics".
Here we keep the word "upwelling," because we are discussing about coastal shallow waters.

>P2, L43: Please replace "input of river water" to "rivers' inputs".
corrected.

>P2, L41,43,44: Please add "s" as follows: water mass changes, terrestrial nutrients' inputs.
added.

>P2, L49-51: Please edit as follows: These short-term pH variations in coastal waters are
>important for local ecosystems as they are mostly caused by natural forcings that have been
>acting before the industrial period, and hence, the local ecosystem is expected to adapt to
>such short-term pH variations as long as they are natural in terms of timing and amplitude.
corrected. Thank you for this nice edition.

>P2, L53: Please edit "changes in land use".
corrected.

>P2, L560: Please remove "Japan" and add "the country", to avoid repetitions.
corrected.

>P2, L563: Please replace "among the stations" by "throughout the stations".
corrected.

>P3, L72: Please replace "in this study" by "Here,".

corrected.

>P3, L76: Please add "in the following stations", after "around the coast of Japan".
added.
>P3, L94: Please add ":" after "three rivers" and remove the word "Rivers" before the
>parentheses.
corrected.

>P3, L95: Please add "s" to nutrient.
corrected.

>P3, L96: Please add a unit after the population numbers here and throughout the manuscript.
>E.g. "residents" could work.
added residents.

>P4, L111: to add always pH units after adding a pH value, e.g., <0.006 pH units. Was this
>difference significant?
The words "pH units" were added to all pH values appeared in the text.
0.006 of pH difference between water intake and settling tank is not significant as this number
is smaller than the estimated uncertainty of monitored pH values after drift corrections
($\pm$0.010 for Mitako, Kashiwazaki, and Ohno and $\pm$0.010 for Shizugawa and Hinase).

>P5, L134: Please remove "was".
removed.

>P6, L179: Please remove the "s" from "numbers".
removed.

>P6, L180: Please edit: "..to one third of what they had before.."
corrected.

>P6, L181: Please replace "diminish" by "reduce".
corrected.

>P6, L186: Please replace "marked" by "remarkable".

>P7, L219: Please replace the "," to ";" before the reference here and throughout the text
>(e.g., L221, 222, etc.).
corrected throughout the text.

>P8, L255: To replace "were hung" with "were installed/fixed".
corrected.

>P9, L274: Do you mean "extremely high"? Please correct.
Sorry for mistyping. It's corrected.

>P9, L274-275: Do you mean that SS variation has a timescale of < 10 days? Please edit.
Here we meant that surface salinity temporally reach to the level of less than 10 salinity unit in these two stations. We slightly modified sentence in the revised text.

>P11, L288: Please add "values" after "raw pH".
Here, we changed the words "raw pH" to "pH after drift correction" following to the comment of another reviewer. So, we don't add "values" in the corresponding point.

>P11, L296: Please add "pH units" after "0.8".
added.

> P11, L296: Please add the "p" in "pCO2" in Italic throughout the text.
Typeface corrected.

>P11, L299: "Phenomena" is not the correct term here. Maybe better to use "patterns".
word changed.

>P11, L296-303: Better to use references in this part.
We added one reference that show artificial diurnal variation of ocean parameters caused by biofouling.

>P11, L318: I suggest to add "concentrations" instead of "values for total alkalinity in the
>entire text (e.g., L329, etc.). And these concentrations "oscillate in a narrow range between
>2222 and 2236...". Please edit.
In the revised manuscript, we stopped to use "Talk values" and just use TA as "alkalinity" itself involves the meaning of "concentration" in its definition.

We revised expression of L318 (now L328-329) in the revised text.

>P13, L325: Which phenomenon are you referring to? If you mean the similarity in S, then
>you can write: This similarity in salinity indicates...
Here, we aimed to indicate about similarity of TA range between northwestern Pacific surface
waters and the "oceanic" end members in Miyako, Kashiwazaki, and Shizugawa. We revised
the sentence to clarify our aim (Line 332).

>P13, L327: What do you mean by "modulated"? maybe use "are not significantly different
>from...".
Sorry for incomprehensible expression here. We changed the words to "different".

>P13, L332-333: Please remove "the fact" and "the" before "the Kuroshio".
corrected.

>P13, L338: Please replace "have already received several modulations by" by "have already
>witnessed several modifications through".
corrected.

>Section 3.3. and Table 1: You are not referring to the methodology used to derive the total
>alkalinity from TA-S relationship. Please note that Jiang et al. (2014) demonstrated that the
>river end-member can only be reliably estimated from this relationship when river input
>dominates as the only significant mechanism controlling the AT-S relationship (e.g., in
>estuaries and river plumes), since physical and biological processes can decouple TA from
>the river water concentration. More information in Hassoun et al. (2019).
>References:
>Jiang, Z.-P., Tyrrell, T., Hydes, D.J., Dai, M., Hartman, S.E., 2014. Variability of alkalinity
>and the alkalinity-salinity relationship in the tropical and subtropical surface ocean. Glob.
>Biogeochem. Cycles 28, 729–742. https://doi.org/10.1002/2013GB004678.
>Hassoun, A.E.R., Fakhri, M., Abboud-Abi Saab, M., Gemayel, E. and De Carlo, E.H., 2019.
>The carbonate system of the Eastern-most Mediterranean Sea, Levantine Sub- basin:
>Variations and drivers. Deep Sea Research Part II: Topical Studies in Oceanography, 164,
>pp.54-73. https://doi.org/10.1016/j.dsr2.2019.03.008 .
We added brief discussion on the validity of simple two end-member mixing model in these
study areas, referring Jiang et al. (2014) (Lines 351-353). We concluded that non-SDC
changes of TA is small in our study areas, as the calculated freshwater endmember in each

area roughly agreed with the observed river TA that flows into each study area (Table 1).

>Section 3.4: Please use the conventional abbreviations of omega calcite ($\Omega$ca) and aragonite
>($\Omega$ar) in the entire paper.
abbreviations corrected.

>P14, L382-383: Please write it as follows: "Since Talk was calculated as a linear equation of
>salinity, it was expected to have a resemblance between the Talk and salinity
>patterns".
corrected.

>P14, L387: What do you mean by "were totally oversaturated with CO2"? Compared to
>what?
We meant to say that observed annual averages of $p\text{CO}_2$ were in equilibrium with, or even
higher than current atmospheric concentration of $\text{CO}_2$. The sentence was revised (Lines 407-
408).

>P16, L400: Please write "Our results indicate...".
corrected.

>P16, L403: Please start the sentence with "The".
corrected.

>P16, L412: Please replace one of the "noted" by "highlighted".
corrected.

>In all the section 4.1, you are referring to your result(s) as phenomenon(a). This is wrong.
>Please edit, and refer to a specific result as a "result", while the phenomenon is usually a
>process that is happening/happened in your study areas, such as OA, eutrophication, etc.
Corrected. Thank you for pointing out this.

>P20, L513-523: Did you check if these annual maximum SDs of the three parameters are
>happening significantly at the same time? You can do ANOVA/ANCOVA tests to check this.
We cannot directly check the simultaneity of the occurrence of maximum SDs of these three
parameters by ANOVA/ANCOVA, because distribution of SD10 against month was not
normal. However, we have checked that not only the month of the occurrence of maximum

SD, seasonal distribution of monthly-averaged SD10 has positive correlation with statistical significance ($r^2 > 0.5$) among these three parameters. In the revised manuscript, we mentioned about this correlation as additional information that support same cause of seasonal variation of SD10 among these parameters (Lines 536-537).

>P24, L606-607: What do you mean by "approximately positive correlation"? It should be
>either positive or negative. Is it significant?
Sorry for incomprehensible sentence. The observed positive correlations were statistically significant, only but data of Shizugawa divided into two groups that follows regression line of Hinase and Ohno. In the revised text we changed this sentence to simply indicate significant positive correlation (Line 640).

>P24, L613: This equation is used according to whom? Please mention the reference (the
>same for the caption of Fig.13b, the purple dashed line).
Here we used classical Redfield ratio of $O_2 : C = 138 : 106$. I added Redfield et al. [1963] as a reference in this sentence.

>P24, L622: What do you mean by "lateral affection"? I'm sure the word "affection" is irrelevant here.
We changed the word to "lateral propagation".

>P26, L626: which analysis are you referring to?
Here we meant to point out our analyses made by Sections 3.2 and 3.4. We changed the sentence in the revised manuscript. (Line 662)

>P26, L629: Please replace "issue" with "aspect".
corrected.

>P27, L651: Do you mean "we determined the nutrients' concentrations..."? Please clarify/edit.
Your understanding is right. We changed the sentence in the revised manuscript. (Lines 688-689)

>P27, L660-667: References?
This speculation comes from our recent observation, and will be published elsewhere. We modified the sentence in the revised text.

>P28, L680: Please replace "derived" by "controlled".
corrected.

>P28, L687: Please replace "acidified" with "acidification".

corrected.

P28, L691-692: How do you know? You didn't discuss this in the paper! I'd suggest to remove this statement.

This is our observation result and are reported in other paper (Fujii et al., 2023). We added this reference (Line 725) .

>P28, L695-696: Any projections studies confirming this?

This projection is also discussed in Fujii et al. (2023). We add this reference in the text (Line 729).

>P28, L696-698: You also didn't tackle this in the paper. Such proposed "solution" needs to be >analyzed taking into consideration various scenarios. Please remove.

We had analyzed in Section 4.3 that the amplitude of the short-term drawdown of pH related to low-salinity events is linear function of liver nutrient concentration, and hence we can say that the short-term drawdown of pH related to low-salinity events will decrease if we can reduce the nutrient concentration of rivers. Similar result has already been reported by Kessouri et al. (2021), so we added this reference in the text (Line 730-732).

On the other hand, I agree that further consideration will be needed to conclude whether such treatment function as the "solution" of future coastal acidification.  We hence deleted the latter half of this sentence.

>Table 1: Please make the last column on the right wider, so units can fit next to the concentrations.

Width corrected.

>What is the source of the atmospheric CO2 number (400 µatm)? Why are you µatm using instead of >ppm?

We corrected this value to 416 ppm, globally averaged atmospheric CO2 concentration at 2021 (WDCGG, 2023).

>I suggest to change the color of one of the stations in this plot as many readers wouldn't tell the >difference.

color changed.

>Please reorder the references from the old to the newest ones (e.g., P2, etc.)

references reordered.

---

## Author Comment (AC2)

Responses to comments from Reviewer 2

Thank you for thorough and constructive comments. Based on the comments from you and another reviewer, we have revised the manuscript as follows:

>1. In the abstract, the author highlighted that this study also revealed the short-term acidification
>occurs more frequently during low-salinity events yet it is not well explained and emphasized in the
>whole manuscript

>2. Figure 7,8, 10 and etc.. The notations on the x-axis are not immediately recognizable to readers.
>Please change to the Year/Month directly. Also, in Figure 7, the notation of 2021.2 = 36.5 days after
>Jan is incorrect.
We have changed notations of X-axis in Figures 7, 8, 10, 11 and 12 to year/mon/date at each graduation. Explanation of X-axis in each figure is changed accordingly.

>3. Should report the pH scale of your pH measurement
Although we have already described in Section 2 that we have used total-scale buffers for calibration of pH sensors, we additionally specified in lines 295-296 that all pH values used in this paper is in total scale.

>4. In Figure 7 and line 271, the salinity change of Miyako Site is covered by the other stations and
>cannot be distinguished.
Y-axis and dot size of Figure 7b are adjusted so that two low peaks of salinity in Miyako Site can barely be distinguished.

>5. Line 271, citations are needed when linking the salinity to the precipitation events in different
>seasons.
A reference for coastal salinity variation and seasonal rainfall in Iwate Prefecture is added in Line 275.

>6. Line 278, it is crucial to provide evidence or citations to support the claim that the low dissolved
>oxygen (DO) observed during the summer is attributed to temperature rather than biological
>processes. Because other factors such as the late stage of algal blooms or other events may also lead
>to reduced DO levels in the water
Dominant contribution of water temperature to DO seasonal variation is proved by using DO(eq) in

Section 4.2. In the revised manuscript we add a notation in Line 281 to refer section 4.2 for more detailed discussion about this statement.

>7. Line 279 – 280 The manuscript should consider the potential influence of salinity changes on >oxygen solubility, particularly in regions like the Ohno Strait with significant salinity variations.
In these lines we refrain discussion about the causes of short-term DO variation and just described the existence of short-term variation. The origin of short-term DO variation is then discussed in In Section 4.2, where we mentioned that, as you pointed out, shot-term variation of salinity played major contribution to the observed DO variation in Hinase and Ohno.

>8. Line 280- 282, in addition, instead of discussing DO, AOU can serve as an appropriate indicator.
In section 4.2 we calculated DO(eq) and DO(diseq), and the later is substantially same variable with AOU. In this paper we use name of DO(diseq) instead of AOU, however, so that we can treat this parameter as a one piece of the biological component package "$C_i$(diseq)," such as pH(diseq) and $\Omega_{ara}$(diseq).

＞9. Line 288, why the raw pH data were used not the pH data after drift correction?
Sorry for imprecise expression. All the numbers plotted in Figure 8 were pH values after drift correction. In the revised manuscript, we corrected expression in line 288 (now line 295) and the legend of Figure 8.

＞10. Figure 9, all the regressions lack of R squared values, RMSE and other statistical analysis details.
RMSE and $r^2$ value are added in each regression equation in Figure 9.

>11.Table 1, it is unclear how the uncertainty terms of the fresh endmember Talk were calculated. >Needs descriptions either in the caption or main text.
Uncertainty terms of freshwater endmember was calculated as standard error of sal = 0 intercept in least squares fitting. In the revised manuscript we add this information in the caption of Table1.

>12. Line 380m the pCO2, the partial pressure should be italic...
Typeface of partial pressure is corrected to italic.

>13. Major comments in Section 4.1 from line 435
>There are numerous questions concerning the comparison between short-term and long-term >variations. Firstly, it is unclear how the standard deviation (SD) was calculated at different time scales.

>For the annual SD, it is not specified whether all data points were used in the calculation. Similarly,
>for the monthly average of 10- days SD, it is unclear whether it was obtained through a moving
>average or only considered the first 10 days, 10-20 days, and 20-30 days of each month. The lack of
>clarity in these definitions and calculations significantly hinders the interpretation of variations, as
>the results heavily depend on how the standard deviations were computed.

Annual average and annual SD were calculated based on the calculated based on the last 1-year data to avoid biases that comes from different time length. We add this information in the caption of Table 2.

$\overline{SD}^m_{10}$ was calculated as the moving average of 10-days SD in each month. We modified the sentence for the explanation of $\overline{SD}^m_{10}$ in lines 455-456.

>14. I do not fully agree that the contribution of the monthly SD to the annual SD can be assessed as
>a ratio or percentage without taking sample size into consideration in the Section 4.1. These need ref
>support.
>Also, if the average monthly SD is smaller than the annual SD, it could be interpreted as the between-
>group variation, implying that monthly mean variations contribute more to the annual variation.
>However, if the average monthly SD is almost the same or even larger than the annual SD, it becomes
>challenging to determine conclusively whether the contribution to the annual variation comes from
>within-group SD (monthly SD) or between-group SD (monthly mean). Again, these discussions
>should have support from statistical literature or equation support.

Following to your comment, we introduced a new variable, annual SD of the monthly averages $(SD_{a<m>})$ in Table 2. contribution of short-term variation with time scale of less than 1 month is then discussed based on the difference between annual SD and $SD_{a<m>}$.

Also. we modified the text of section 4.1 to refrain from discussing about mechanism of temporal variation (e.g., temperature-driven vs biological-driven) at this stage, limiting our discussion to the relative contribution of variations with different time scales.

>15. Line 593. This statement requires further clarification. The thermodynamic process, which
>involves changes in temperature and other constants, also encompasses the air-sea gas exchange
>process. The equilibrium term mentioned in the manuscript is the combined outcome of these two
>processes. It is essential to note that the equilibrium term is often considered negligible because the
>effects of these two processes tend to counterbalance each other.

Please note that we defined $C_i(eq)$ as the estimated concentration of parameter $i$ in equilibrium with the current atmosphere under the observed water temperature and salinity. Variation of $C_i(eq)$ hence only represents purely thermodynamic changes, and non-equilibrium term caused by insufficient gas exchange is incorporated in $C_i(diseq)$ if exist.

In this sentence, we already know that variation $C_i$(eq) is less than 10 % of that of $C_i$(diseq). As variation of non-equilibrium term caused by insufficient gas exchange is less than that of $C_i$(eq) by definition, we can conclude that variation of $C_i$(diseq) caused by insufficient gas exchange is also negligible without any discussion.

>16. Lines 601-605. The contribution of the SD to the variation of pH should be carefully interpreted.
>Merely relying on the correlation of the SD may not conclusively indicate that pH variations are
>solely contributed by other factors. For example, a situation might arise where high pH variation
>(SDm 10) occurs in the first 5 days of a month, while high salinity variation occurs in the last five
>days of a ten-day period. In such cases, the correlation of the SD might not fully capture the
>relationship between the two variables. To ensure robust and meaningful conclusions, a statistical
>study and appropriate justification should accompany the use of the correlation of the SD.
In lines 601-605 in the original paper, we evaluated relative contribution of non-thermodynamic component to the observed annual variation based not on the correlation between SDs but on the seasonal amplitude of non-thermodynamic component that can be detected both from Table 4 and Figures 11 - 12.

---

## Referee Report (RR1)

Responses to comments from Reviewer 1

Thank you for thorough and constructive comments. Based on the comments from you and anotherreviewer, we have revised the manuscript as follows:

>P2, L63-65: You are mentioning that "The observed range of pH trend within the Japanese
>coast corresponds to 85% of that observed in 83 coastal systems in the world (Carstensen
>and Duarte, 2019), and this result suggests that the Japanese coastal area can function as
>the "sample shelf" of the coastal environment for the entire world." I guess you mean that
>"The observed range of pH trend within the Japanese coast is in harmony with 85% of the
>ones observed in 83 coastal systems around the world (Carstensen and Duarte, 2019)".
>Right? But this does not mean that the Japanese coastal area can represent THE word's
>coastal environment. However, it can be considered as one of the coastal areas that can be
>used as a "sample shelf" or "coastal ocean acidification test area".
In this sentence we meant to point out that the observed range of pH trend within the Japanese coast is equivalent to 85% of that observed in 83 coastal systems in the world. Also, I agree with your comment that Japanese coasts will be useful as a sample shelf of coastal acidification but not that of coastal environment itself. I modified the words in this paragraphto reflect your comments (lines 62-64).
Please replace "this result suggests that the Japanese coastal area can function as the "sample shelf" of the coastal acidification in the world." to "this result suggests that the Japanese coastal area can be considered as a "sample shelf" for coastal acidification studies.".

>In P2, L60-63, you have mentioned that "The Japan Ministry of the Environment (MOE)
>conducts regular pH monitoring at over 2,000 coastal stations around Japan from the early
>1980s until the present, and the obtained data showed significant variability in the multi-
>decadal pH trend from −0.012 y−1 to +0.009 y−1 among the stations" and in P3, L76-81,
>you mentioned the 5 stations without justifying the reason you chose them among 2000
>stations! It would be very useful to do so before describing each station in details. Monitoring cites in this study was not selected from the MOE stations but newly launched bytwo recent pH monitoring programs (Study of Biological Effects of Acidification and Hypoxia (BEACH) and Ocean Acidification Adaptation Project (OAAP)). The reason for determination of monitoring locations differed by program: in BEACH, two stations were set to represent "natural" state of Japanese coastal environment with relatively low anthropogenic nutrient loadings. In OAAP, three stations were selected from major farming areas of pacific oyster. We add these information in the revised text (lines 79-82).

>Also, the time range of data is different between these 5 stations. Why didn't you choose the
>ones that have exactly the same time range of beginning and end of measurements to

>facilitate the inter-comparison?

As these stations were launched in order by two different programs, we couldn't start/end years of these stations. Please extenuate that this is the first synthesis effort of OA monitoring stations operated by different founders / programs.

Please mention that this is "the 1st synthesis effort of OA monitoring stations operated by different founers/programs" clearly in the text.

>P4, L111: You've mentioned that "All sensors were replaced every 2 months". Despite

>calibration? Why?

Several sensors including pH has only three-months lifetime of their batteries, and hence we changed all sensors every two months for safety. In the revised manuscript we add this information (Line 116).

>P4: You didn't mention the precision and accuracy of measurements for any parameter.

>Why? This is crucial to add.

Precision of each parameter is added in the revised text. (Lines 108-110).

>P4: You used CO2sys Excel v2.1 (Lewis and Wallace, 1998) but the correct reference should

>be (Pierrot and Wallace, 2006): Pierrot, D.E. Lewis, and D.W.R. Wallace. 2006. MS Excel

>Program Developed for CO2 System Calculations. ORNL/CDIAC-105a. Carbon Dioxide

>Information Analysis Center, Oak Ridge National Laboratory, U.S. Department of Energy,

>Oak Ridge, Tennessee. doi: 10.3334/CDIAC/otg.CO2SYS_XLS_CDIAC105a ; Index of
>/ftp/co2sys/CO2SYS_calc_XLS_v2.1        (lbl.gov).
Thank you for your comment. I modified the reference.

It is (Pierrot and Wallace, 2006) not (Pierrot et al, 2006).

>P4: Also, you didn't mention the uncertainty of your carbonate system measurements. This

>can be easily calculated through the same CO2sys Excel with the new features added by Orr

>et al. (2018).

Measurement precision of DIC and Talk reported from JAMSTEC was added in Line 129. Uncertainty of carbon-derived parameters (pCO2 and □ara) are function of T, S, and carbon parameters (pH and Talk in this case), and estimated uncertainty of these parameters calculated by CO2sys were added in Section 3.4 (Lines391-400).

>You use the term "eutrophicated" in the paper. I would definitely suggest to use the
>conventional terms "eutrophic, oligotrophic, etc.".
We changed the term "eutrophicated" to "eutrophic".

>Referring to total alkalinity as Talk in the paper is very confusing. Better to type in its

>conventional abbreviation "T$_A$".

Talk is a traditional expression of total alkalinity in the field of ocean carbon monitoring, and current major programs such as GO-SHIP still use this expression. However, I agree that T$_A$ is more used in the field of ocean acidification study, and I changed abbreviation of this parameter to T$_A$ in the revised manuscript.

Please edit in Figure 10 and line 401.

>Sections 3.3. and 3.4.: You have to mention the r2 and the significance of each
>equation/relationship in each station. It would be good to add them in a separate table of in
>the plots (Figure 9).

The r$^2$ value and RMSE were added in each equation listed in Figure 9.

>Figure 7: You are showing the years in a decimal way which might be confusing for readers.
>I would suggest to add the months and the year abbreviation, e.g., Jan-20, Jan-21, etc.
>

$_A$ll x-axis in Figures 7, 8, 10, 11, and 12 were changed to show date/month/year instead of years in a decimal.

The new format is "year.month", and is different in Figure 8(b). I still find this confusing.

I would suggest to add the month than the abbreviated year, e.g. Jan-20 or Dec.21, etc.

>In general, the discussion needs more references, particularly for the statements used to
>explain the results, the methodology of using SDs and plotting those against each other. $_A$lso,
>this section still needs more comparison with other studies from the area and beyond.

We added several references to compare our results with former studies; such as

*annual pH ranges (Lines 318-320) / Please add "for" before the Baltic Sea.

*low buffer effect in low-salinity water (Lines 403-404) / Please re-arrange the references from old to new.

*DIC vs DO relationship (Lines 649-650) / Please edit: "in the coastal area of East China Sea.."

*effect of reduction of nutrient loadings to diminish short-term drawdown of $\square_{ar}$ (Lines 730-732)

>P1, L20: The forcings mentioned here are already changing due to climate change, and not
> "can change". Please edit accordingly.

We changed the words to "...forcings are changing..."

>P1, L25,27: What do you mean by "anthropogenic loadings"? Could be replaced by
>"anthropogenic pressures" that encompass all types of atmospheric and terrestrial
>discharges and emissions generated by anthropogenic activities and infrastructures.

Here we meant "anthropogenic nutrient loadings," but "anthropogenic pressures" are better. In the revised text we changed these words accordingly.

Then please clarify that you mean anthropogenic nutrients' loadings in L680, 683, 710, 711, 734.

>P2, L38: The sentence is not correct, as the ocean is not lowering its pH per say. Please edit

>this sentence "The ocean is lowering its pH because of anthropogenic CO2 input..." to "The

>ocean is witnessing a reduction in its pH due to anthropogenic CO2 sequestration..." Sentence corrected.

>P2, L40: Please remove "also" after pH and edit as follows: the pH shows... corrected.

>P2, L42: It's not only coastal upwelling but also the downwelling that is taking atmospheric

>CO2 from surface layers to the deep ocean. I would suggest to replace it with "coastal

>dynamics".

Here we keep the word "upwelling," because we are discussing about coastal shallow waters.

>P2, L43: Please replace "input of river water" to "rivers' inputs". corrected.

>P2, L41,43,44: Please add "s" as follows: water mass changes, terrestrial nutrients' inputs. added.

>P2, L49-51: Please edit as follows: These short-term pH variations in coastal waters are

>important for local ecosystems as they are mostly caused by natural forcings that have been

>acting before the industrial period, and hence, the local ecosystem is expected to adapt to

>such short-term pH variations as long as they are natural in terms of timing and amplitude. corrected. Thank you for this nice edition.

>P2, L53: Please edit "changes in land use". corrected.

>P2, L560: Please remove "Japan" and add "the country", to avoid repetitions. corrected.

Please edit: "The country's Ministry of…".

>P2, L563: Please replace "among the stations" by "throughout the stations". corrected.

>P3, L72: Please replace "in this study" by "Here,".

corrected.

>P3, L76: Please add "in the following stations", after "around the coast of Japan".added.

>P3, L94: Please add ":" after "three rivers" and remove the word "Rivers" before the
>parentheses.
corrected.

>P3, L95: Please add "s" to nutrient.
corrected.

>P3, L96: Please add a unit after the population numbers here and throughout the manuscript.
>E.g. "residents" could work.
added residents.

>P4, L111: to add always pH units after adding a pH value, e.g., <0.006 pH units. Was this

>difference significant?

The words "pH units" were added to all pH values appeared in the text.
0.006 of pH difference between water intake and settling tank is not significant as this number is
smaller than the estimated uncertainty of monitored pH values after drift corrections
(±0.010 for Mitako, Kashiwazaki, and Ohno and ±0.010 for Shizugawa and Hinase).

>P5, L134: Please remove "was".
removed.

>P6, L179: Please remove the "s" from "numbers".
removed.

>P6, L180: Please edit: "..to one third of what they had before.."
corrected.

>P6, L181: Please replace "diminish" by "reduce".
corrected.

>P6, L186: Please replace "marked" by "remarkable".

>P7, L219: Please replace the "," to ";" before the reference here and throughout the text
>(e.g., L221, 222, etc.).
corrected throughout the text.

>P8, L255: To replace "were hung" with "were installed/fixed". corrected.

>P9, L274: Do you mean "extremely high"? Please correct.
Sorry for mistyping. It's corrected.

>P9, L274-275: Do you mean that SS variation has a timescale of < 10 days? Please edit.
Here we meant that surface salinity temporally reach to the level of less than 10 salinity unit in these two stations. We slightly modified sentence in the revised text.

>P11, L288: Please add "values" after "raw pH".
Here, we changed the words "raw pH" to "pH after drift correction" following to the comment of another reviewer. So, we don't add "values" in the corresponding point.

>P11, L296: Please add "pH units" after "0.8". added.

> P11, L296: Please add the "p" in "pCO2" in Italic throughout the text. Typeface corrected.

>P11, L299: "Phenomena" is not the correct term here. Maybe better to use "patterns".
word changed.

>P11, L296-303: Better to use references in this part.
We added one reference that show artificial diurnal variation of ocean parameters caused by biofouling.

>P11, L318: I suggest to add "concentrations" instead of "values for total alkalinity in the
>entire text (e.g., L329, etc.). And these concentrations "oscillate in a narrow range between
>2222 and 2236...". Please edit.
In the revised manuscript, we stopped to use "Talk values" and just use TA as "alkalinity" itself involves the meaning of "concentration" in its definition.

We revised expression of L318 (now L328-329) in the revised text.

>P13, L325: Which phenomenon are you referring to? If you mean the similarity in S, then

>you can write: This similarity in salinity indicates...

Here, we aimed to indicate about similarity of TA range between northwestern Pacific surface waters and the "oceanic" end members in Miyako, Kashiwazaki, and Shizugawa. We revised the sentence to clarify our aim (Line 332).

>P13, L327: What do you mean by "modulated"? maybe use "are not significantly different

>from...".

Sorry for incomprehensible expression here. We changed the words to "different".

>P13, L332-333: Please remove "the fact" and "the" before "the Kuroshio".
corrected.

>P13, L338: Please replace "have already received several modulations by" by "have already

>witnessed several modifications through".corrected.

>Section 3.3. and Table 1: You are not referring to the methodology used to derive the total

>alkalinity from TA -S relationship. Please note that Jiang et al. (2014) demonstrated that the

>river end-member can only be reliably estimated from this relationship when river input

>dominates as the only significant mechanism controlling the TA-S relationship (e.g., in

>estuaries and river plumes), since physical and biological processes can decouple TA from

>the river water concentration. More information in Hassoun et al. (2019).

>References:

>Jiang, Z.-P., Tyrrell, T., Hydes, D.J., Dai, M., Hartman, S.E., 2014. Variability of alkalinity

>and the alkalinity-salinity relationship in the tropical and subtropical surface ocean. Glob.

>Biogeochem. Cycles 28, 729–742. https://doi.org/10.1002/2013GB004678.

>Hassoun, A.E.R., Fakhri, M., Abboud-Abi Saab, M., Gemayel, E. and De Carlo, E.H., 2019.

>The carbonate system of the Eastern-most Mediterranean Sea, Levantine Sub- basin:

>Variations and drivers. Deep Sea Research Part II: Topical Studies in Oceanography, 164,

>pp.54-73. https://doi.org/10.1016/j.dsr2.2019.03.008 .

We added brief discussion on the validity of simple two end-member mixing model in these study areas, referring Jiang et al. (2014) (Lines 351-353). We concluded that non-SDC changes of TA is small in our study areas, as the calculated freshwater endmember in each

area roughly agreed with the observed river TA that flows into each study area (Table 1).

Please edit: "In most study areas, calculated freshwater endmember of TA-S regressions agreed with the observed TA of one of freshwater sources (Table 1), indicating that non-dilution/concentration changes of TA (Jiang et al. 2014) was relatively small in these areas, and…"

>Section 3.4: Please use the conventional abbreviations of omega calcite (Ωca) and aragonite
>(Ωar) in the entire paper.abbreviations
corrected.

>P14, L382-383: Please write it as follows: "Since Talk was calculated as a linear equation of
>salinity, it was expected to have a resemblance between the Talk and salinity
>patterns".corrected.

>P14, L387: What do you mean by "were totally oversaturated with CO2"? Compared to
>what?
We meant to say that observed annual averages of $p\text{CO}_2$ were in equilibrium with, or even higher than current atmospheric concentration of $\text{CO}_2$. The sentence was revised (Lines 407-408).

Please edit: WDCGG, 2023

>P16, L400: Please write "Our results indicate...".
corrected.

>P16, L403: Please start the sentence with "The".
corrected.

>P16, L412: Please replace one of the "noted" by "highlighted". corrected.

>In all the section 4.1, you are referring to your result(s) as phenomenon(a). This is wrong.
>Please edit, and refer to a specific result as a "result", while the phenomenon is usually a
>process that is happening/happened in your study areas, such as $O_2$, eutrophication, etc.
Corrected. Thank you for pointing out this.

Please edit: "Such a result can be attributed to the perturbation of salinity that…"

>P20, L513-523: Did you check if these annual maximum SDs of the three parameters are
>happening significantly at the same time? You can do ANOVA/ANCOVA tests to check this.
We cannot directly check the simultaneity of the occurrence of maximum SDs of these three parameters by ANOVA/ANCOVA, because distribution of SD10 against month was not normal. However, we have checked that not only the month of the occurrence of maximum

SD, seasonal distribution of monthly-averaged SD10 has positive correlation with statistical significance ($r^2 > 0.5$) among these three parameters. In the revised manuscript, we mentioned about this correlation as additional information that support same cause of seasonal variation of SD10 among these parameters (Lines 536-537).

Please edit: "variations with shorter time scale than one month contribute to some parts of the annual variation in the case of salinity."

>P24, L606-607: What do you mean by "approximately positive correlation"? It should be

>either positive or negative. Is it significant?

Sorry for incomprehensible sentence. The observed positive correlations were statistically significant, only but data of Shizugawa divided into two groups that follows regression line of Hinase and Ohno. In the revised text we changed this sentence to simply indicate significant positive correlation (Line 640).

>P24, L613: This equation is used according to whom? Please mention the reference (the

>same for the caption of Fig.13b, the purple dashed line).

Here we used classical Redfield ratio of $O_2 : C = 138 : 106$. I added Redfield et al. [1963] as

a reference in this sentence.

>P24, L622: What do you mean by "lateral affection"? I'm sure the word "affection" is

irrelevant here.

We changed the word to "lateral propagation".

>P26, L626: which analysis are you referring to?

Here we meant to point out our analyses made by Sections 3.2 and 3.4. We changed

the sentence in the revised manuscript. (Line 662)

>P26, L629: Please replace "issue" with

"aspect".

corrected.

>P27, L651: Do you mean "we determined the nutrients' concentrations..."? Please

clarify/edit.

Your understanding is right. We changed the sentence in the revised manuscript. (Lines

688-689)

>P27, L660-667: References?

This speculation comes from our recent observation, and will be published elsewhere.

We modified the sentence in the revised text.

Please edit: (unpublished data).

>P28, L680: Please replace "derived" by

"controlled".

corrected.

>P28, L687: Please replace "acidified" with "acidification".

corrected.

P28, L691-692: How do you know? You didn't discuss this in the paper! I'd suggest to remove this statement.

This is our observation result and are reported in other paper (Fujii et al., 2023). We added this reference (Line 725) .

>P28, L695-696: Any projections studies confirming this?

This projection is also discussed in Fujii et al. (2023). We add this reference in the text (Line 729).

>P28, L696-698: You also didn't tackle this in the paper. Such proposed "solution" needs to be analyzed taking into consideration various scenarios. Please remove.

We had analyzed in Section 4.3 that the amplitude of the short-term drawdown of pH related to low-salinity events is linear function of liver nutrient concentration, and hence we can say that the short-term drawdown of pH related to low-salinity events will decrease if we can reduce the nutrient concentration of rivers. Similar result has already been reported by Kessouri et al. (2021), so we added this reference in the text (Line 730-732).

On the other hand, I agree that further consideration will be needed to conclude whether such treatment function as the "solution" of future coastal acidification. We hence deleted the latter half of this sentence.

>Table 1: Please make the last column on the right wider, so units can fit next to the concentrations.

Width corrected.

>What is the source of the atmospheric CO2 number (400 μatm)? Why are you μatm using instead of ppm?

We corrected this value to 416 ppm, globally averaged atmospheric CO2 concentration at 2021(WDCGG, 2023).

>I suggest to change the color of one of the stations in this plot as many readers wouldn't tell the difference.

color changed.

>Please reorder the references from the old to the newest ones (e.g., P2, etc.)

references reordered.

In the caption of Table 2, please edit: "…biases that come from different time length."

P27, L673: Please edit "…, in Hinase and Ohno,"

The discussion is still weak. Results can be better compared with regional and global studies. In addition, many of the discussed statements in the conclusion could be used in the discussion section, while keeping the conclusion section more focused on results and concise.

---

## Editor Decision (ED1)

**Referee #1**

Thanks for the authors for addressing my previous comments. The ms. has definitely improved a lot. I still have a couple of minor comments and I still think that the discussion could be better enriched with other references and deeper comparisons.

Responses to comments from Reviewer 1

Thank you for thorough and constructive comments. Based on the comments from you and anotherreviewer, we have revised the manuscript as follows:

>P2, L63-65: You are mentioning that "The observed range of pH trend within the Japanese
>coast corresponds to 85% of that observed in 83 coastal systems in the world (Carstensen
>and Duarte, 2019), and this result suggests that the Japanese coastal area can function as
>the "sample shelf" of the coastal environment for the entire world." I guess you mean that
>"The observed range of pH trend within the Japanese coast is in harmony with 85% of the
>ones observed in 83 coastal systems around the world (Carstensen and Duarte, 2019)".
>Right? But this does not mean that the Japanese coastal area can represent THE word's
>coastal environment. However, it can be considered as one of the coastal areas that can be
>used as a "sample shelf" or "coastal ocean acidification test area".
In this sentence we meant to point out that the observed range of pH trend within the
Japanese coast is equivalent to 85% of that observed in 83 coastal systems in the world. Also,
I agree with your comment that Japanese coasts will be useful as a sample shelf of coastal
acidification but not that of coastal environment itself. I modified the words in this paragraphto
reflect your comments (lines 62-64).

Please replace "this result suggests that the Japanese coastal area can function as the
"sample shelf" of the coastal acidification in the world." to "this result suggests that the
Japanese coastal area can be considered as a "sample shelf" for coastal acidification
studies.".

>In P2, L60-63, you have mentioned that "The Japan Ministry of the Environment (MOE)
>conducts regular pH monitoring at over 2,000 coastal stations around Japan from the early
>1980s until the present, and the obtained data showed significant variability in the multi-
>decadal pH trend from −0.012 y−1 to +0.009 y−1 among the stations" and in P3, L76-81,
>you mentioned the 5 stations without justifying the reason you chose them among 2000
>stations! It would be very useful to do so before describing each station in details. Monitoring
cites in this study was not selected from the MOE stations but newly launched bytwo recent
pH monitoring programs (Study of Biological Effects of Acidification and Hypoxia (BEACH)
and Ocean Acidification Adaptation Project (OAAP)). The reason for determination of
monitoring locations differed by program: in BEACH, two stations were set to represent
"natural" state of Japanese coastal environment with relatively low anthropogenic nutrient
loadings. In OAAP, three stations were selected from major farming areas of pacific oyster.
We add these information in the revised text (lines 79-82).

> Also, the time range of data is different between these 5 stations. Why didn't you choose the
>ones that have exactly the same time range of beginning and end of measurements to

>facilitate the inter-comparison?

⚨s these stations were launched in order by two different programs, we couldn't start/end years of these stations. Please extenuate that this is the first synthesis effort of O⚨ monitoring stations operated by different founders / programs.

Please mention that this is "the 1st synthesis effort of OA monitoring stations operated by different founers/programs" clearly in the text.

>P4, L111: You've mentioned that "⚨ll sensors were replaced every 2 months". Despite

>calibration? Why?

Several sensors including pH has only three-months lifetime of their batteries, and hence we changed all sensors every two months for safety. In the revised manuscript we add this information (Line 116).

>P4: You didn't mention the precision and accuracy of measurements for any parameter.

>Why? This is crucial to add.

Precision of each parameter is added in the revised text. (Lines 108-110).

>P4: You used CO2sys Excel v2.1 (Lewis and Wallace, 1998) but the correct reference should

>be (Pierrot and Wallace, 2006): Pierrot, D.E. Lewis, and D.W.R. Wallace. 2006. MS Excel

>Program Developed for CO2 System Calculations. ORNL/CDI⚨C-105a. Carbon Dioxide

>Information ⚨nalysis Center, Oak Ridge National Laboratory, U.S. Department of Energy,

>Oak Ridge, Tennessee. doi: 10.3334/CDI⚨C/otg.CO2SYS_XLS_CDI⚨C105a ; Index of

>/ftp/co2sys/CO2SYS_calc_XLS_v2.1        (lbl.gov).

Thank you for your comment. I modified the reference.

It is (Pierrot and Wallace, 2006) not (Pierrot et al, 2006).

>P4: ⚨lso, you didn't mention the uncertainty of your carbonate system measurements. This

>can be easily calculated through the same CO2sys Excel with the new features added by Orr

>et al. (2018).

Measurement precision of DIC and Talk reported from J⚨MSTEC was added in Line 129. Uncertainty of carbon-derived parameters (pCO2 and □ara) are function of T, S, and carbon parameters (pH and Talk in this case), and estimated uncertainty of these parameters calculated by CO2sys were added in Section 3.4 (Lines391-400).

>You use the term "eutrophicated" in the paper. I would definitely suggest to use the
>conventional terms "eutrophic, oligotrophic, etc.".
We changed the term "eutrophicated" to "eutrophic".

>Referring to total alkalinity as Talk in the paper is very confusing. Better to type in its

>conventional abbreviation "T♙".

Talk is a traditional expression of total alkalinity in the field of ocean carbon monitoring, and current major programs such as GO-SHIP still use this expression. However, I agree that T♙ is more used in the field of ocean acidification study, and I changed abbreviation of this parameter to T♙ in the revised manuscript.

Please edit in Figure 10 and line 401.

>Sections 3.3. and 3.4.: You have to mention the r2 and the significance of each

>equation/relationship in each station. It would be good to add them in a separate table of in

>the plots (Figure 9).

The $r^2$ value and RMSE were added in each equation listed in Figure 9.

>Figure 7: You are showing the years in a decimal way which might be confusing for readers.

>I would suggest to add the months and the year abbreviation, e.g., Jan-20, Jan-21, etc.

>

♙ll x-axis in Figures 7, 8, 10, 11, and 12 were changed to show date/month/year instead of years in a decimal.

The new format is "year.month", and is different in Figure 8(b). I still find this confusing.

I would suggest to add the month than the abbreviated year, e.g. Jan-20 or Dec.21, etc.

>In general, the discussion needs more references, particularly for the statements used to

>explain the results, the methodology of using SDs and plotting those against each other. ♙lso,
>this section still needs more comparison with other studies from the area and beyond.
We added several references to compare our results with former studies; such as

*annual pH ranges (Lines 318-320) / Please add "for" before the Baltic Sea.

*low buffer effect in low-salinity water (Lines 403-404) / Please re-arrange the references from old

to new.

*DIC vs DO relationship (Lines 649-650) / Please edit: "in the coastal area of East China Sea.."
*effect of reduction of nutrient loadings to diminish short-term drawdown of □ar (Lines 730-732)

>P1, L20: The forcings mentioned here are already changing due to climate change, and not

> "can change". Please edit accordingly.

We changed the words to "...forcings are changing..."

>P1, L25,27: What do you mean by "anthropogenic loadings"? Could be replaced by

>"anthropogenic pressures" that encompass all types of atmospheric and terrestrial

>discharges and emissions generated by anthropogenic activities and infrastructures.
Here we meant "anthropogenic nutrient loadings," but "anthropogenic pressures" are better.
In the revised text we changed these words accordingly.

Then please clarify that you mean anthropogenic nutrients' loadings in L680, 683, 710, 711, 734.

>P2, L38: The sentence is not correct, as the ocean is not lowering its pH per say. Please edit

>this sentence "The ocean is lowering its pH because of anthropogenic CO2 input..." to "The

>ocean is witnessing a reduction in its pH due to anthropogenic CO2 sequestration..."
Sentence corrected.

>P2, L40: Please remove "also" after pH and edit as follows: the pH shows...
corrected.

>P2, L42: It's not only coastal upwelling but also the downwelling that is taking atmospheric

>CO2 from surface layers to the deep ocean. I would suggest to replace it with "coastal

>dynamics".
Here we keep the word "upwelling," because we are discussing about coastal shallow waters.

>P2, L43: Please replace "input of river water" to "rivers' inputs".
corrected.

>P2, L41,43,44: Please add "s" as follows: water mass changes, terrestrial nutrients' inputs.
added.

>P2, L49-51: Please edit as follows: These short-term pH variations in coastal waters are

>important for local ecosystems as they are mostly caused by natural forcings that have been

>acting before the industrial period, and hence, the local ecosystem is expected to adapt to
>such short-term pH variations as long as they are natural in terms of timing and amplitude.
corrected. Thank you for this nice edition.

>P2, L53: Please edit "changes in land use".
corrected.

>P2, L560: Please remove "Japan" and add "the country", to avoid repetitions.
corrected.
Please edit: "The country's Ministry of…".

>P2, L563: Please replace "among the stations" by "throughout the stations".
corrected.

>P3, L72: Please replace "in this study" by "Here,".

corrected.

>P3, L76: Please add "in the following stations", after "around the coast of Japan".added.

>P3, L94: Please add ":" after "three rivers" and remove the word "Rivers" before the
>parentheses.
corrected.

>P3, L95: Please add "s" to nutrient.
corrected.

>P3, L96: Please add a unit after the population numbers here and throughout the manuscript.
>E.g. "residents" could work.
added residents.

>P4, L111: to add always pH units after adding a pH value, e.g., <0.006 pH units. Was this

>difference significant?

The words "pH units" were added to all pH values appeared in the text.
0.006 of pH difference between water intake and settling tank is not significant as this number is
smaller than the estimated uncertainty of monitored pH values after drift corrections
(±0.010 for Mitako, Kashiwazaki, and Ohno and ±0.010 for Shizugawa and Hinase).

>P5, L134: Please remove "was".
removed.

>P6, L179: Please remove the "s" from "numbers".
removed.

>P6, L180: Please edit: "..to one third of what they had before.."
corrected.

>P6, L181: Please replace "diminish" by "reduce".
corrected.

>P6, L186: Please replace "marked" by "remarkable".

>P7, L219: Please replace the "," to ";" before the reference here and throughout the text
>(e.g.,    L221,    222,    etc.).
corrected throughout the text.

>P8, L255: To replace "were hung" with "were installed/fixed". corrected.

>P9, L274: Do you mean "extremely high"? Please correct.
Sorry for mistyping. It's corrected.

>P9, L274-275: Do you mean that SS variation has a timescale of < 10 days? Please edit.
Here we meant that surface salinity temporally reach to the level of less than 10 salinity unit in these two stations. We slightly modified sentence in the revised text.

>P11, L288: Please add "values" after "raw pH".
Here, we changed the words "raw pH" to "pH after drift correction" following to the comment of another reviewer. So, we don't add "values" in the corresponding point.

>P11, L296: Please add "pH units" after "0.8". added.

> P11, L296: Please add the "p" in "pCO2" in Italic throughout the text. Typeface corrected.

>P11, L299: "Phenomena" is not the correct term here. Maybe better to use "patterns".
word changed.

>P11, L296-303: Better to use references in this part.
We added one reference that show artificial diurnal variation of ocean parameters caused by biofouling.

>P11, L318: I suggest to add "concentrations" instead of "values for total alkalinity in the

>entire text (e.g., L329, etc.). And these concentrations "oscillate in a narrow range between

>2222 and 2236...". Please edit.
In the revised manuscript, we stopped to use "Talk values" and just use TA as "alkalinity" itself involves the meaning of "concentration" in its definition.

We revised expression of L318 (now L328-329) in the revised text.

>P13, L325: Which phenomenon are you referring to? If you mean the similarity in S, then

>you can write: This similarity in salinity indicates...

Here, we aimed to indicate about similarity of T♟ range between northwestern Pacific surface waters and the "oceanic" end members in Miyako, Kashiwazaki, and Shizugawa. We revised the sentence to clarify our aim (Line 332).

>P13, L327: What do you mean by "modulated"? maybe use "are not significantly different

>from...".

Sorry for incomprehensible expression here. We changed the words to "different".

>P13, L332-333: Please remove "the fact" and "the" before "the Kuroshio".
corrected.

>P13, L338: Please replace "have already received several modulations by" by "have already

>witnessed several modifications through".corrected.

>Section 3.3. and Table 1: You are not referring to the methodology used to derive the total

>alkalinity from T♟ -S relationship. Please note that Jiang et al. (2014) demonstrated that the

>river end-member can only be reliably estimated from this relationship when river input

>dominates as the only significant mechanism controlling the ♟T-S relationship (e.g., in

>estuaries and river plumes), since physical and biological processes can decouple T♟ from

>the river water concentration. More information in Hassoun et al. (2019).

>References:

>Jiang, Z.-P., Tyrrell, T., Hydes, D.J., Dai, M., Hartman, S.E., 2014. Variability of alkalinity

>and the alkalinity-salinity relationship in the tropical and subtropical surface ocean. Glob.

>Biogeochem. Cycles 28, 729–742. https://doi.org/10.1002/2013GB004678.

>Hassoun, ♟.E.R., Fakhri, M., ♟bboud-♟bi Saab, M., Gemayel, E. and De Carlo, E.H., 2019.

>The carbonate system of the Eastern-most Mediterranean Sea, Levantine Sub- basin:

>Variations and drivers. Deep Sea Research Part II: Topical Studies in Oceanography, 164,

>pp.54-73. https://doi.org/10.1016/j.dsr2.2019.03.008 .

We added brief discussion on the validity of simple two end-member mixing model in these study areas, referring Jiang et al. (2014) (Lines 351-353). We concluded that non-SDC changes of T♟ is small in our study areas, as the calculated freshwater endmember in each

area roughly agreed with the observed river TA that flows into each study area (Table 1).

Please edit: "In most study areas, calculated freshwater endmember of TA-S regressions agreed with the observed TA of one of freshwater sources (Table 1), indicating that non-dilution/concentration changes of TA (Jiang et al. 2014) was relatively small in these areas, and…"

>Section 3.4: Please use the conventional abbreviations of omega calcite ($\Omega_{ca}$) and aragonite
>($\Omega_{ar}$) in the entire paper.abbreviations
corrected.

>P14, L382-383: Please write it as follows: "Since Talk was calculated as a linear equation of
>salinity, it was expected to have a resemblance between the Talk and salinity
>patterns".corrected.

>P14, L387: What do you mean by "were totally oversaturated with CO2"? Compared to
>what?
We meant to say that observed annual averages of $p$CO$_2$ were in equilibrium with, or even higher than current atmospheric concentration of CO$_2$. The sentence was revised (Lines 407-408).

Please edit:  WDCGG, 2023

>P16, L400: Please write "Our results indicate...".
corrected.

>P16, L403: Please start the sentence with "The".
corrected.

>P16, L412: Please replace one of the "noted" by "highlighted". corrected.

>In all the section 4.1, you are referring to your result(s) as phenomenon(a). This is wrong.

>Please edit, and refer to a specific result as a "result", while the phenomenon is usually a
>process that is happening/happened in your study areas, such as OA, eutrophication, etc.
Corrected. Thank you for pointing out this.

Please edit: "Such a result can be attributed to the perturbation of salinity that…"

>P20, L513-523: Did you check if these annual maximum SDs of the three parameters are
>happening significantly at the same time? You can do ANOVA/ANCOVA tests to check this.
We cannot directly check the simultaneity of the occurrence of maximum SDs of these three parameters by ANOVA/ANCOVA, because distribution of SD10 against month was not normal. However, we have checked that not only the month of the occurrence of maximum

SD, seasonal distribution of monthly-averaged SD10 has positive correlation with statistical significance ($r^2 > 0.5$) among these three parameters. In the revised manuscript, we mentioned about this correlation as additional information that support same cause of seasonal variation of SD10 among these parameters (Lines 536-537).

Please edit: "variations with shorter time scale than one month contribute to some parts of the annual variation in the case of salinity."

>P24, L606-607: What do you mean by "approximately positive correlation"? It should be

>either positive or negative. Is it significant?

Sorry for incomprehensible sentence. The observed positive correlations were statistically significant, only but data of Shizugawa divided into two groups that follows regression line of Hinase and Ohno. In the revised text we changed this sentence to simply indicate significant positive correlation (Line 640).

>P24, L613: This equation is used according to whom? Please mention the reference (the

>same for the caption of Fig.13b, the purple dashed line).

Here we used classical Redfield ratio of $O_2 : C = 138 : 106$. I added Redfield et al. [1963] as

a reference in this sentence.

>P24, L622: What do you mean by "lateral affection"? I'm sure the word "affection" is

irrelevant here.

We changed the word to "lateral propagation".

>P26, L626: which analysis are you referring to?

Here we meant to point out our analyses made by Sections 3.2 and 3.4. We changed

the sentence in the revised manuscript. (Line 662)

>P26, L629: Please replace "issue" with

"aspect".

corrected.

>P27, L651: Do you mean "we determined the nutrients' concentrations..."? Please

clarify/edit.

Your understanding is right. We changed the sentence in the revised manuscript. (Lines

688-689)

>P27, L660-667: References?

This speculation comes from our recent observation, and will be published elsewhere.

We modified the sentence in the revised text.

Please edit: (unpublished data).

>P28, L680: Please replace "derived" by

"controlled".

corrected.

>P28, L687: Please replace "acidified" with "acidification".

corrected.

P28, L691-692: How do you know? You didn't discuss this in the paper! I'd suggest to remove this statement.

This is our observation result and are reported in other paper (Fujii et al., 2023). We added this reference (Line 725) .

>P28, L695-696: Any projections studies confirming this?

This projection is also discussed in Fujii et al. (2023). We add this reference in the text (Line 729).

>P28, L696-698: You also didn't tackle this in the paper. Such proposed "solution" needs to be analyzed taking into consideration various scenarios. Please remove.

We had analyzed in Section 4.3 that the amplitude of the short-term drawdown of pH related to low-salinity events is linear function of liver nutrient concentration, and hence we can say that the short-term drawdown of pH related to low-salinity events will decrease if we can reduce the nutrient concentration of rivers. Similar result has already been reported by Kessouri et al. (2021), so we added this reference in the text (Line 730-732).
On the other hand, I agree that further consideration will be needed to conclude whether such treatment function as the "solution" of future coastal acidification. We hence deleted the latter half of this sentence.

>Table 1: Please make the last column on the right wider, so units can fit next to the concentrations.

Width corrected.

>What is the source of the atmospheric CO2 number (400 μatm)? Why are you μatm using instead of ppm?

We corrected this value to 416 ppm, globally averaged atmospheric CO2 concentration at 2021(WDCGG, 2023).

>I suggest to change the color of one of the stations in this plot as many readers wouldn't tell the difference.
color changed.

>Please reorder the references from the old to the newest ones (e.g., P2, etc.)

references reordered.

In the caption of Table 2, please edit: "…biases that come from different time length."

P27, L673: Please edit "…, in Hinase and Ohno,"

The discussion is still weak. Results can be better compared with regional and global studies. In addition, many of the discussed statements in the conclusion could be used in the discussion section, while keeping the conclusion section more focused on results and concise.

Referee #2

Overall, this revised version demonstrates significant improvements in clarifying key results and discussions. However, there remain several notable issues that require attention:

The figures and captions appear unchanged despite the author's response indicating revisions were made.

We acknowledge and appreciate the authors' efforts in introducing a new term, which adds clarity. However, it's essential to ensure that this term is adequately described and utilized in the discussion within Section 4.2 for comprehensive understanding.

The use of units, specifically 'umol kg-1' or 'umol/kg,' lacks consistency throughout the manuscript.

While there have been improvements, the overall writing quality of the manuscript still requires further refinement. Continued efforts in this regard would contribute to a more polished final product.
* * *
Referee #3

This paper presents the findings from continuous measurements conducted at five monitoring sites established along the coast of Japan. The short-period variations in pH are categorized into physical and biochemical factors.
The paper has already undergone initial peer review, and I have read the reports of the two reviewers, the author's responses to them, and the revised manuscript after the first peer review. The data presented are rare in terms of continuous observations conducted annually, and this paper, which provides data from as many as five stations, holds significant value. Therefore, I recommend the publication of this paper in Biogeosciences. While no major revisions are necessary since the paper has undergone peer review once, I suggest some modifications to enhance the comprehensibility of the content for readers.
Firstly, I found the first reviewer's comments regarding the method for calculating standard deviations across the three time scales (10-days, month and year) and their comparative discussion somewhat challenging to grasp at first glance. Although I eventually comprehended the concept after multiple readings, to enhance reader comprehension without requiring additional effort, I suggest including an illustrative example figure. Attached in the PDF is an example displaying the average temperature in Tokyo for the year 2022 The upper panel exhibits the annual variation, while the lower panel displays the January variation. These are supplemented by visual representations of each standard deviation. Providing similar figures using water temperature data from one of the stations as an example would greatly facilitate understanding of the discussion. Please note that the example figure I created is for illustrative purposes and need not match the exact figure used in the paper.

(Line 549) property > parameter?

(Line 556) It is easier to understand if you describe that DOdiseq is AOU multiplied by -1.

(Line 559) mol > mole?

(Line 569) The notation "DICmin" is used, but this may be misleading since the DIC calculated from pHmin is not smaller, but rather larger. The term "pHmin" is introduced near Line 313. If it is explicitly stated after this sentence that "pHmin" is to be used for subsequent pH calculations. "pHmin" and "DIC" can be referred to as "pH" and "DIC," respectively.

(Line 652) You mention that the respiration quotient in the sediment is possible cause for a larger DIC/O2 ratio larger the Redfield ratio. However, I believe there is another consideration to be made. The Redfield ratio of 0.77 assumes that both DIC and O2 are no longer being exchanged with the atmosphere. Given that oxygen is gradually released into the atmosphere over time, it is reasonable to expect that the DIC/O2 ratio should be significantly higher. The author should know that air-sea difference in pCO2 can deviate significantly from zero while AOU approaches zero at the surface of the open ocean. If biological activity is occurring at a distance from the station, rather than in its immediate vicinity, it is plausible that the DIC/O2 ratio could be inflated due to the release or absorption of oxygen into the atmosphere during the advection period.

(Line 675 and Figure 14) I do not agree with the theory that the slope changes with a difference in salinity greater than 1.0. This is because 1.0 salinity has no scientific significance. How is the slope of each area if not divided by salinity 1.0? According to Fig. 14, it may be obvious that the greater the anthropogenic load, the greater the short-term variation of pH in response to the short-term variation of salinity.

(Line 681) the dose > those?

(Figure 7, 8, 10, 11 and 12) The horizontal axis should be corrected to a month-year notation, such as Jan-21, Jan-22. This is an item that was also pointed out by two reviewers in the first peer review.

[Figure]

---

## Author Response (AR2)

Responses to reviewer 1

Thanks again for your various constructive suggestions/comments in second revision. Based on your comments, we modified the manuscript as follows:

>Please replace "this result suggests that the Japanese coastal area can function as the
>"sample shelf" of the coastal acidification in the world." to "this result suggests that the
>Japanese coastal area can be considered as a "sample shelf" for coastal acidification
>studies."
=>Sentence revised.  (L64-65)

>Please mention that this is "the 1st synthesis effort of OA monitoring stations operated by
>different founers/programs" clearly in the text.
=>We inserted this sentence in line70-72, with slight modification in the following sentence.

> It is (Pierrot and Wallace, 2006) not (Pierrot et al, 2006).
>
=>Reference corrected. (lines 131-132)

>Please edit in Figure 10 and line 401.
>
=>corrected.

>The new format is "year.month", and is different in Figure 8(b). I still find this confusing. I would
>suggest to add the month than the abbreviated year, e.g. Jan-20 or Dec.21, etc.
>
=>As we use 1.5-years data from 2020/5/26 to 2021/12/31, we cannot omit year from the X-
axes. Instead, we changed the style of X-axes of all time series to that of Fig.8(b).

>Please add "for" before the Baltic Sea.
>
=>added. (Line 323)

>Please re-arrange the references from old to new.
>
=>corrected. (Lines 403-404)

>Please edit: "in the coastal area of East China Sea.."
>
=>corrected. (Line 671)

>Then please clarify that you mean anthropogenic nutrients' loadings in L680, 683, 710, 711,
>734.
>
=>We modified these sentences to clarify that we are discussing about anthropogenic nutrients'
loadings.

>Please edit: "The country's Ministry of...".
>
=>corrected. (L60)

>Please edit: "In most study areas, calculated freshwater endmember of TA-S regressions
>agreed with the observed TA of one of freshwater sources (Table 1), indicating that non-
>dilution/concentration changes of TA (Jiang et al. 2014) was relatively small in these areas,
>and..."
>
=>corrected. (Lines 355-357)

>Please edit: WDCGG, 2023
>
=>corrected. (Line 405)

>Please edit: "Such a result can be attributed to the perturbation of salinity that..."
>
=>corrected. (Line 477)

>Please edit: "variations with shorter time scale than one month contribute to some parts of the
>annual variation in the case of salinity."
>
=>Here, we hold our original expression as your modification changes the meaning of the
sentence significantly. We believe that we had been able to show the probability of the same
cause of seasonal variation of SD10 among DO, pHmin and Wara.

>Please edit: (unpublished data).

>

=>corrected. (Line 742)

>In the caption of Table 2, please edit: "...biases that come from different time length."

>

=>corrected.

>P27, L673: Please edit "..., in Hinase and Ohno,"

>

=>corrected.

>The discussion is still weak. Results can be better compared with regional and global studies.

>In addition, many of the discussed statements in the conclusion could be used in the

>discussion section, while keeping the conclusion section more focused on results and concise.

>

=>We added several sentences (e.g., Lines 279-280, 556-567) so that the readers can compare the observed variabilities of each parameter in this study with that of former studies.

Responses to reviewer 2

Thanks again for your various constructive suggestions/comments in second revision.
All time-series figures are now replaced to new versions. Also, all notations of $\mu$mol/kg remained in the 1st revision were replaced to $\mu$mol kg$^{-1}$.

Responses to reviewer 3

Thanks again for your various constructive suggestions/comments in second revision.
In response to your comments on the complexity of SDs in our analyses, we re-edited the names of various SDs to have consistency in their naming strategy (Lines 452-460, Section 4.1 and 4.2). Also, we added a schematic explanation of SDs in Appendix1.
Other responses to the comments are as follows:

>(Line 549) property > parameter?
>
=>corrected. (now Line 571)

>(Line 556) It is easier to understand if you describe that DOdiseq is AOU multiplied by -1.
>
=>thank you for this idea. I added this description into Line 578.

>(Line 559) mol > mole?
>
=>corrected. (now Line 581)

>(Line 569) The notation "DICmin" is used, but this may be misleading since the DIC >calculated from pHmin is not smaller, but rather larger. The term "pHmin" is >introduced near Line 313. If it is explicitly stated after this sentence that "pHmin" is to >be used for subsequent pH calculations. "pHmin" and "DIC" can be referred to as "pH" >and "DIC," respectively.
>
=>Here, we had misspelled DICmax to DICmin. Thank you for this notification.

>(Line 652) You mention that the respiration quotient in the sediment is possible cause for a larger >DIC/O2 ratio larger the Redfield ratio. However, I believe there is another consideration to be made. >The Redfield ratio of 0.77 assumes that both DIC and O2 are no longer being exchanged with the >atmosphere. Given that oxygen is gradually released into the atmosphere over time, it is reasonable to >expect that the DIC/O2 ratio should be significantly higher. The author should know that air-sea >difference in pCO2 can deviate significantly from zero while AOU approaches zero at the surface of the >open ocean. If biological activity is occurring at a distance from the station, rather than in its immediate

>vicinity, it is plausible that the DIC/O2 ratio could be inflated due to the release or absorption of oxygen

>into the atmosphere during the advection period.

>

=>Thank you for important suggestion. We agree that this process cannot be rejected without any expranation, and hence we added description on how we conclude that this process is not main cause of the observed high − $\Delta DIC_{max}(diseq)/\Delta DO(diseq)$ ratio in Hinase area (lines 676 - 689). The largest reason is that we cannot observe high − $\Delta DIC_{max}(diseq)/\Delta DO(diseq)$ ratio in Ohno, while monthly wind speed is rather higher in this place than in Hinase.

>(Line 675 and Figure 14) I do not agree with the theory that the slope changes with a difference in

>salinity greater than 1.0. This is because 1.0 salinity has no scientific significance. How is the slope of

>each area if not divided by salinity 1.0? According to Fig. 14, it may be obvious that the greater the

>anthropogenic load, the greater the short-term variation of pH in response to the short-term variation of

>salinity.

>

=> Generally, suspended particle export of the river increases linearly with that of river discharge, while percentage of particle organic materials (POM) within suspended particles decreases along with the increase of suspended particle export (e.g., Coynel et al., 2005; Point et al., 2007; Zhang et al., 2013; Bukaveckas 2022). As this result, increasing rate of the POM transport with that of river discharge tends to become weak when river discharge becomes high enough (Coynel et al., 2005; Kim et al., 2020). If we assume that short term pH variation is mainly caused by the degradation of POM discharged from rivers into coastal areas and subsequential biological production by using nutrients released from the degraded POM, then we can expect that $\Delta avgSD_{10}{}^{m}(pH)$ / $\Delta avgSD_{10}{}^{m}(sal)$ will also become small at the months with high river discharge (i.e., high $avgSD_{10}{}^{m}(sal)$). We added this explanation in the revised text (Lines 717-725).

Although there is certain reason why we apply two-lines regression, we have no specific reason why the boundary of two regression lines become at the $avgSD_{10}{}^{m}(sal)$ of 1.0. We determined this boundary so that correlation coefficient of both lines become maximum, so there may be some undetermined relationship between this value and low/high discharge boundary in Japanese rivers. even so, we cannot clearly explain such relationship so far. We, therefore, changed Fig. 14 to show both all-data regression and high-$avgSD_{10}{}^{m}(sal)$ data regression. Fig.15 has also changed to show these two results.

>(Line 681) the dose > those?

>

=>corrected. (now Line 728)

>(Figure 7, 8, 10, 11 and 12) The horizontal axis should be corrected to a month-year notation, such as

>Jan-21, Jan-22. This is an item that was also pointed out by two reviewers in the first peer review.

>

=>Sorry that we somehow failed to convert these figures to new ones. Now these figures are renewed.

---

## Author Response (AR3)

Thanks again for various constructive suggestions/comments from both of you. Based on your comments, we modified the manuscript as follows:

Responses to reviewer 1

> (Line 57 and more) No space between "20" and "degree-N" is required.
>
Our English proofreading service requested this space, but now we re-checked BG manuscript guidelines and found that the proper formulation of coordinates in BG is "a degree sign and a space when naming the direction (e.g. 30° N, 25° E)".
In the revised manuscript, we corrected styles of all coordinates in the text.

> (Line 136) The saturation state of calcite may not be necessary as the paper only mentions that of
> aragonite.
>
We removed the words regarding $\Omega_{ca}$.

> (Line 190) "to > 1 mg L-1" > "by > 1mg L-1"
>
corrected.

> (Line 220 and more) "141,000 tN y-1" > "1.41 × 10^5 tN y−1" Significant figures should be aligned
> with other places.
>
Numbers corrected to be displayed with index.
Figures will be properly aligned by editorial office at the publication.

> (Line 268 and more) "amplitude was the highest and ~ the lowest" > "amplitude was the largest ~
> and the smallest". The adjective for "amplitude" is generally considered to be "large" rather than
> "high".
>
We corrected the expressions.

> (Line 429) Waldbusser et al. [2015] > Waldbusser et al. (2015)
>
corrected.

> (Line 496 in Table 2) Ohono > Ohno

>

corrected.

Responses to reviewer 2

> P27, L679: Please write "seawater" in one word.

>

corrected.

> P27, L681: Please correct "also".

>

corrected.

> P27, L687: Please write "..not the main cause.."

>

corrected.